# LEARNING HIERARCHICAL IMAGE SEGMENTATION FOR RECOGNITION AND BY RECOGNITION

**Tsung-Wei Ke**[*1]  **Sangwoo Mo**[*2]  **Stella X. Yu**[1,2]
[1]University of California, Berkeley  [2]University of Michigan, Ann Arbor
{twke,stellayu}@berkeley.edu {swmo,stellayu}@umich.edu

## ABSTRACT

Large vision and language models learned directly through image-text associations often lack detailed visual substantiation, whereas image segmentation tasks are treated separately from recognition, supervisedly learned without interconnections.

Our key observation is that, while an image can be recognized in multiple ways, each has a consistent part-and-whole visual organization. Segmentation thus should be treated not as an end task to be mastered through supervised learning, but as an internal process that evolves with and supports the ultimate goal of recognition.

We propose to integrate a hierarchical segmenter into the recognition process, *train* and *adapt* the entire model solely on image-level recognition objectives. We learn hierarchical segmentation *for free* alongside recognition, automatically uncovering part-to-whole relationships that not only underpin but also enhance recognition.

Enhancing the Vision Transformer (ViT) with adaptive segment tokens and graph pooling, our model surpasses ViT in unsupervised part-whole discovery, semantic segmentation, image classification, and efficiency. Notably, our model (trained on *unlabeled* 1M ImageNet images) outperforms SAM (trained on 11M images and 1 billion masks) by absolute 8% in mIoU on PartImageNet object segmentation.

## 1 INTRODUCTION

Learning visual recognition through the association of images with textual descriptions, as demonstrated by CLIP (Radford et al., 2021) and GPT-4 (Achiam et al., 2023), has achieved significant success. However, direct supervision that links images to semantics often fails to provide *what-is-where* visual substantiation. For instance, a model might be trained to categorize the image in Fig. 1 as *ink*, *girl*, or *woman* without understanding *how* these categories differ in this image. Similarly, visual segmentation can be trained using masks at multiple granularities (Kirillov et al., 2023), but such models may not grasp *how* segments are related to each other and to overall image recognition!

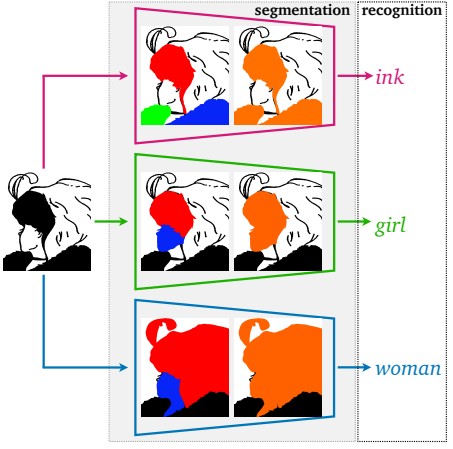

Figure 1: **Our insight is that image segmentation and recognition form a visual parsing continuum, and their consistency is more essential than individual text labels for recognition.** We may recognize this image as *ink*, *girl*, or *woman*. While the foreground (colored areas) may vary, it always has a consistent hierarchical segmentation: *three individual blobs* when *no person* is recognized, or *parts* (face, hair) of the *person* recognized as *girl* or *woman*. Instead of treating segmentation and recognition as separate tasks, we model them concurrently by *including segmentation in the loop for recognition*. With recognition objectives *solely at the image level*, not only can hierarchical segmentation be learned *for free*, but better and substantiated recognition also arises from such internal part-to-whole consistency.

---

[*]Equal contribution. Code available at https://github.com/twke18/CAST.

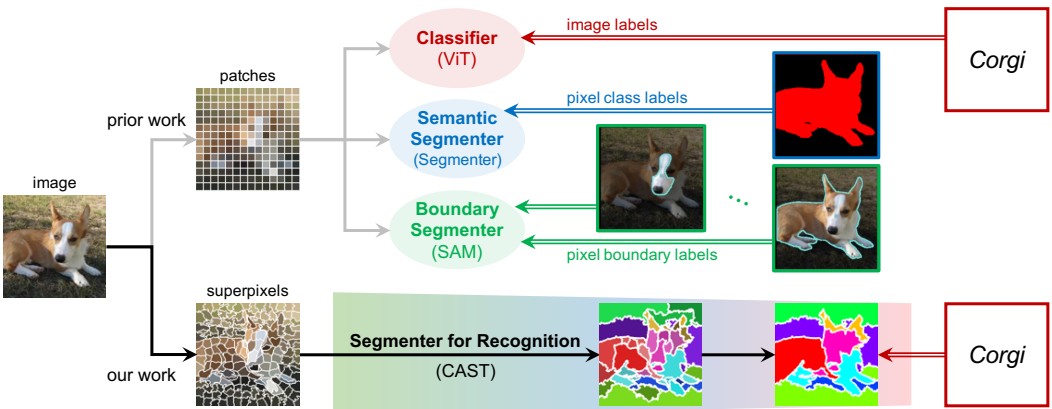

Figure 2: **While prior work uses patches as visual units and treats segmentation and recognition as separate, supervised tasks with distinct models and data, our work uses superpixels as visual units and integrates hierarchical segmentation into the recognition process, learning it internally from a single recognition objective.** Classifiers like ViT (Dosovitskiy et al., 2020) learn recognition from image-level labels. Semantic Segmenters such as Segmenter (Strudel et al., 2021) learn object segments from pixel-level class labels but lack part-whole granularity. Boundary Segmenters like SAM (Kirillov et al., 2023) learns regions of multiple granularities from boundary labels without hierarchical organization. In contrast, our **Segmenter for Recognition** (CAST) integrates a fine-to-coarse segment hierarchy directly into the recognition process. By graph-pooling over segment tokens, it effectively solves all three tasks concurrently within a visual parsing continuum.

**Our first insight** is that segmentation and recognition form a continuum of visual parsing, substantiating concepts not primarily through textual labels, but through visual organization. For the image in Fig. 1: **1)** Recognition of *ink* is concurrent with organizing black pixels as *three individual blobs* which form *ink group*; **2)** Recognition of *girl* is concurrent with organizing *profile face* and *black hair* which form *girl's head*; **3)** Recognition of *woman* is concurrent with organizing *three-quarter face* and *black hair* which form *woman's head*. Recognition of the *whole* is validated by its segmentation into *parts*. There is always a *part* segmentation hierarchy consistent with the *whole* recognition, and each varies in conjunction with the other. We cannot simultaneously recognize the image as *girl* and the *nose* of the *woman*. In other words, the actual semantics of recognition may *not* be crucial, but the *concurrency and consistency* between segmentation and recognition *are*.

In human vision, composition of distinctive parts could facilitate scene understanding (Biederman, 1987), whereas coarse recognition could help explain connected parts (Maurer et al., 2002). Information flows both ways between parts and wholes, clarifying each other in a final consistent percept (Tanaka & Farah, 1993; Tanaka & Simonyi, 2016; Tanaka et al., 2019).

In computer vision, prior works (Fig. 2 top) treat segmentation and recognition tasks separately, each having its own dedicated model and annotated training data. **1)** Recognition models are trained using image-level labels that distinguish between categories (Deng et al., 2009) or instances (Wu et al., 2018). Each image is encoded into a global feature vector (He et al., 2016; Dosovitskiy et al., 2020), which typically emphasizes the most discriminative parts for recognition (Selvaraju et al., 2017). **2)** Segmentation models are trained using pixel-level labels (Long et al., 2015; Kirillov et al., 2023), employing skip connections (Lin et al., 2017; Cheng et al., 2021) to propagate information across various spatial resolutions and coverage areas. **3)** Due to their distinct architectural designs, recognition models cannot be directly used for segmentation tasks. Instead, architectural modifications and fine-tuning with segmentation labels are often required (Ahn & Kwak, 2018).

**Our second insight** is that, with segmentation and recognition on a visual parsing continuum, segmentation should be treated *not* as an end task to be mastered through external supervised learning, but an *internal* process that evolves with and supports the ultimate goal of recognition.

We propose to integrate hierarchical segmentation into the recognition process (Fig. 2 bottom), to naturally ensure that our recognition model achieves the desired consistency and concurrency between segmentation and recognition. Specifically, our model processes the input image through a fine-to-coarse segmentation that correlates with part-to-whole relationships, culminating in a global feature vector that encapsulates the entire image. This segmentation, internal to the recognition process and not an end goal in itself, ultimately enhances recognition with visual spatial parsing.

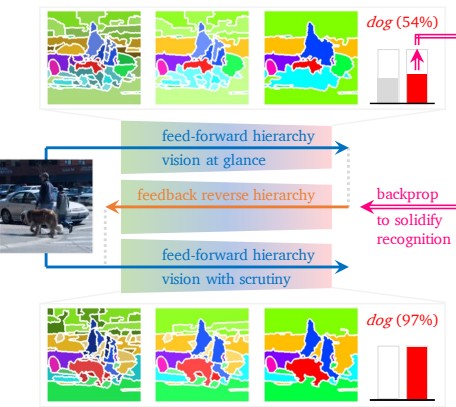

Figure 3: **Our model performs segmentation and recognition simultaneously during test-time adaptation: Initial predictions in a feedforward hierarchy capture vision at a glance, whereas enhancements in a reverse hierarchy captures vision with scrutiny.** It processes an image of a *dog, human, and car* in a feed-forward hierarchy, initially recognizing the *dog* with 54% activation based on only the *back* of the *dog*. After backpropagating to increase *dog* activation, the model undergoes test-time adaptation in a reverse hierarchy. This adjustment allows the next feed-forward process to uncover the *whole dog* and boost *dog* activation to 97%! Our segmentation and recognition thus mutually influence and enhance each other.

Consequently, our model can be directly optimize for recognition using the final global feature vector, while developing hierarchical segmentation *for free* that spatially grounds recognition in the image.

Our concept can be encapsulated by the phrase: **Segmentation Of Recognition, By Recognition, and For Recognition.** "**Of**" refers to looping segmentation in the process of recognition. "**By**" indicates that the learning process of internal segmentation is driven by image-level recognition objectives, without any segment-level supervision. "**For**" reflects that the outcomes of our model automatically uncover part-to-whole relationships that not only ground but also enhance recognition.

We implement our concept by innovating Vision Transformer (ViT) (Dosovitskiy et al., 2020) on two aspects. **1)** We use arbitrarily-shaped superpixels instead of square patches on a regular grid as the visual units for ViT tokens. **2)** We use graph-pooling to group these segment tokens successively towards recognition, forming a fine-to-coarse segmentation hierarchy that reflects part-to-whole relationships. The entire model is learned solely from image recognition objectives, either unsupervised (Wu et al., 2018; He et al., 2020) or supervised (Touvron et al., 2021). Our model is abbreviated as CAST, signifying that it Concurrently learns segmentation and recognition using Adaptive Segment Tokens. Unlike the traditional text-inspired Vision Transformer, which utilizes regular patches as *visual words*, our vision-inspired CAST employs superpixels that adhere to visual contours as *visual words*. In this sense, our CAST embodies a true *vision* transformer model.

By looping hierarchical segmentation into recognition, CAST delivers four major results. **1)** CAST derives a hierarchical segmentation by grouping segment tokens from fine to coarse. SAM (Kirillov et al., 2023), trained with 11 million images and 1 billion masks for multi-scale segmentations, fails to grasp part-to-whole relationships and produce hierarchical segmentation. **2)** CAST learns segmentation *for free* directly from an image recognition objective. During training, our model adapts internal segmentation to optimize the final image recognition. During testing, it processes images in a feed-forward hierarchy, making initial predictions that capture *vision at a glance* (Ahissar et al., 2009). For uncertain recognition, our model can continue test-time adaptation (TTA) (Sun et al., 2020) to solidify recognition. With targeted feedback backpropagating in a reverse hierarchy, it refines internal part-to-whole segmentation alongside improvements in final recognition, capturing *vision with scrutiny* (Fig. 3 and Appx. C). **3)** CAST simultaneously performs segmentation and recognition, matching or surpassing previous methods like HSG (Ke et al., 2022) for hierarchical segmentation and Swin Transformer (Liu et al., 2021) for classification. While these methods require specially designed architectures for each task, CAST efficiently manages both using a single unified model. **4)** CAST represents a natural evolution in vision transformer design, utilizing superpixels instead of square patches as visual units. This approach allows our model to achieve more accurate segmentation compared to patch-based ViT. It excels across multiple tasks, including both unsupervised and supervised semantic segmentation as well as attention-based figure-ground segmentation.

## 2 RELATED WORKS

**Concurrent segmentation and recognition** was explored before the advent of deep learning. Previously, models simultaneously performed recognition by grouping compatible patches and segmentation by grouping visually similar pixels through detected pixel-patch relations. This approach resulted in object-specific (Yu et al., 2002; Yu & Shi, 2003b) and figure-ground segmentation (Maire, 2010; Maire et al., 2011). However, these methods relied on manually crafted features and pre-trained object part detectors. In contrast, our model is entirely data-driven and trained from scratch.

**Vision transformers** have undergone significant advancements (Han et al., 2022) since the introduction of ViT (Dosovitskiy et al., 2020). Two primary strategies exist for enhancing efficiency: One reduces the number of tokens through spatial pooling, leveraging principles from hierarchical convolution (Liu et al., 2021; Heo et al., 2021; Dong et al., 2022; Ma et al., 2023), and the other assesses token significance to selectively prune tokens (Goyal et al., 2020; Rao et al., 2021; Marin et al., 2021; Zeng et al., 2022; Bolya et al., 2023). Our approach diverges from these methods in two fundamental ways: **1)** Our token pooling is designed to support a segmentation-recognition continuum within a visual parsing hierarchy; efficiency emerges only as a beneficial byproduct. **2)** Unlike any other ViT model, our model produces a consistent hierarchical segmentation.

**Hierarchical image segmentation** aims to group pixels consistently across multiple granularities. A representative approach is agglomerative clustering (Sharon et al., 2006; Arbelaez et al., 2010), which starts by extracting pixel features, initializing clusters, and then merging them based on feature similarity. Recent works often adopt a supervised approach that uses top-down decomposition to detect coarse semantic instances and break them down into finer semantic parts (de Geus et al., 2021; Wei et al., 2024; Li et al., 2022b;a). To circumvent part annotations, self-supervised learning is used to enhance part segmentation for novel categories through cross-image pixel correspondence (Sun et al., 2023) or feature clustering (Pan et al., 2023). Another line of approaches predicts part- and object-level segmentations separately (Li et al., 2023; Wang et al., 2024; Qi et al., 2024), which often leads to misaligned segmentation across granularities. In contrast, our work modernizes the agglomerative approach by integrating **1)** representation learning, **2)** hierarchical segmentation, and **3)** image recognition within a transformer architecture.

**Additional related works** on superpixels and clustering methods can be found in Appx. D.

## 3 OUR HIERARCHICAL SEGMENTER AND RECOGNIZER ON A CONTINUUM

Our goal is to integrate hierarchical segmentation and recognition into a continuous visual parsing framework, where the segmenter is internal to and supports the recognition process (Fig. 2). During training, the segmenter is developed with the recognizer, both guided solely by the image-level recognition objective. During testing, optimizing the recognizer by maximizing its winning activations results in backpropagation that refines the segmenter (Fig. 3). In our concurrent segmentation and recognition framework called CAST, segmentation not only substantiates recognition but is also directed by it, ensuring that both processes mutually enhance each other.

We begin by utilizing the widely adopted ViT for recognition tasks and identify two major challenges in adapting it for hierarchical segmentation (Fig. 4): **1)** The use of fixed-shape patches leads to poor characterization of complex visual contours (Bolya et al., 2022); **2)** The absence of explicit segmentation makes it difficult to enforce consistent pixel grouping across granularities.

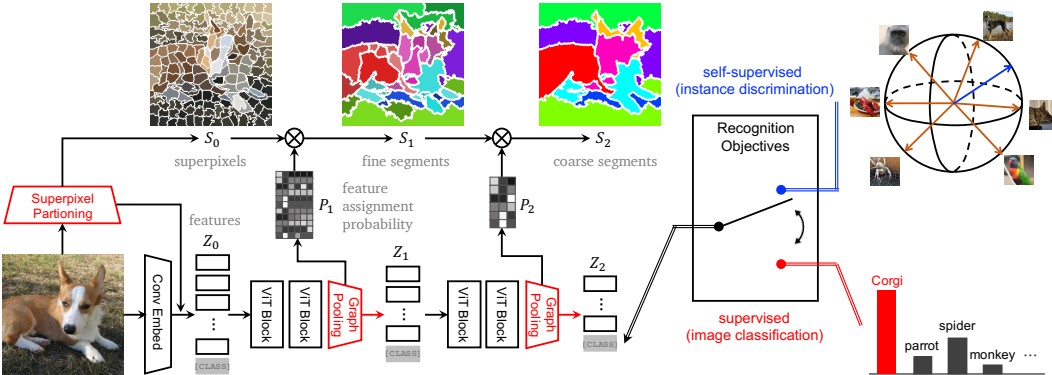

Figure 4: **Our model implements our concept of concurrency and consistency in visual parsing by innovating ViT with adaptive segment tokens and progressive graph pooling.** It starts with superpixels instead of square patches, and applies graph pooling to merge fine segments $S_{l-1}$ into coarse segments $S_l$. Both segment transition probability $P_l$ and segment feature $Z_l$ are learned to optimize an image-level recognition objective, which could be self-supervised instance discrimination or supervised image classification. Without any external supervision, we uncover object wholes (*dog*) along with small details (*ears*) and thin structures (*legs*), validating the effectiveness of our concept.

Our solution is **1)** to replace fixed-shape patches with adaptive segments that adhere to visual contours, and **2)** to employ graph pooling over segments to progressively construct a parsing hierarchy. In Fig. 4, our model utilizes segments as visual units and alternates between transformer encoder blocks that extract segment features and graph pooling modules that merge finer segments into coarser ones. We describe adaptive segments and graph pooling below, and list our algorithm in Appx. E.1.

### 3.1 Adaptive Segment Tokens from Superpixels

Segments consist of arbitrary groups of pixels. For effective hierarchical segmentation, we require representations for both the pixel groupings of segments, labeled $S_0, S_1, S_2, \ldots$, and the segment features, labeled $Z_0, Z_1, Z_2, \ldots$, across levels from fine to coarse.

Our initial segmentation, $S_0$, uses superpixels that match the low-level features within areas and align with visual contours. We achieve this using SEEDS (Bergh et al., 2012), which breaks the image into color-consistent, locally connected regions. Discussions on the choice of superpixel methods can be found in Appx A.2. We then combine these superpixels into larger segments, forming a segmentation hierarchy with precisely outlined contours, as shown in Fig. 4 top.

Our initial segment feature, $Z_0$, uses both visual and spatial features. We first apply convolutional layers to the input image to generate pixel-level features, labeled $X_{\text{cnn}}$. The visual feature for each superpixel, $X_s$, is the average of pixel features $X_{\text{cnn}}$ within that superpixel. Similarly, the positional encoding for each superpixel, $E_{\text{pos}}$, is the average of pixel positional encodings, which are set at the same resolution as $X_{\text{cnn}}$, within that superpixel. The initial segment feature, $Z_0$, is the sum of these two superpixel features appended with the usual class token $X_{\text{class}}$ for ViT: $Z_0 = [X_s + E_{\text{pos}}; X_{\text{class}}]$.

Starting from superpixels in $S_0$, we group these finest segment tokens $Z_0$ into $L$ levels of coarser segment tokens $(Z_1, \ldots, Z_L)$ to progressively capture more global visual contexts. These adaptive segment tokens enable direct derivation of image segmentation, $(S_1, \ldots, S_L)$, from the segment index of each pixel, eliminating the need for post-processing. Our approach contrasts with previous methods such as SegSort (Hwang et al., 2019) and HSG (Ke et al., 2022), which maintain separation between image segmentation and feature extraction across their entire models.

### 3.2 Graph Pooling for Hierarchical Segmentation

We construct a hierarchical segmentation by applying graph pooling to adaptive segment tokens, progressively aggregating them from fine to coarse levels. The aggregation process utilizes a soft assignment probability, $P_l$, which maps a fine segment $a$ at level $l - 1$ to a coarser segment $c$ at level $l$. We initialize coarse segments $Z_l$, dimensioned by segment and feature channel counts, by sampling centroids of fine segment tokens $Z_{l-1}$. The closer segment $a$ is to segment $c$ in the feature space, as measured by feature similarity function sim, the larger the probability $P_l$ of assigning $a$ to $c$:

$$P_l = \text{Prob}(a \to c) \propto \text{sim}(Z_{l-1}[a], Z_l[c]). \tag{1}$$

Representing initial segmentation $S_0$ in a binary partition matrix, dimensioned by pixel and segment counts, we derive a segmentation hierarchy based on progressive segment membership transitions:

$$S_l = S_{l-1} \times P_l, \quad l = 1, 2, \ldots, L. \tag{2}$$

While $S_{l-1}$ is binary, $S_l$ is a soft segmentation that can be hardened through a common winner-take-all strategy. Segment tokens $Z_l$ are updated according to $S_l$ by averaging $Z_{l-1}$ using $P_l$ and adding them with an MLP head to the previously initialized values: with ./ denoting element-wise division,

$$Z_l \Leftarrow Z_l + \text{MLP}(P_l^\top Z_{l-1}./P_l^\top 1). \tag{3}$$

To demonstrate how our superpixels and hierarchical segmenter enhance recognition, we compare our CAST to ViT, which uses patches and maintains the same number of tokens throughout the model. For the sake of comparisons, we also derive a hierarchical segmentation from *the already trained* ViT tokens by applying K-means clustering in a consistent fine-to-coarse manner (details in Appx. E.5). Fig.5 demonstrates that not only the use of superpixels allows our segmentation to follow visual contours more closely, but enforcing segmentation consistency across granularities also enables CAST to preserve small details and thin structures in coarse segmentation. Consequently, while ViT and CAST are trained on identical *unlabeled* data, CAST significantly excels in uncovering whole objects without any supervision, validating the benefits of our superpixels and token pooling.

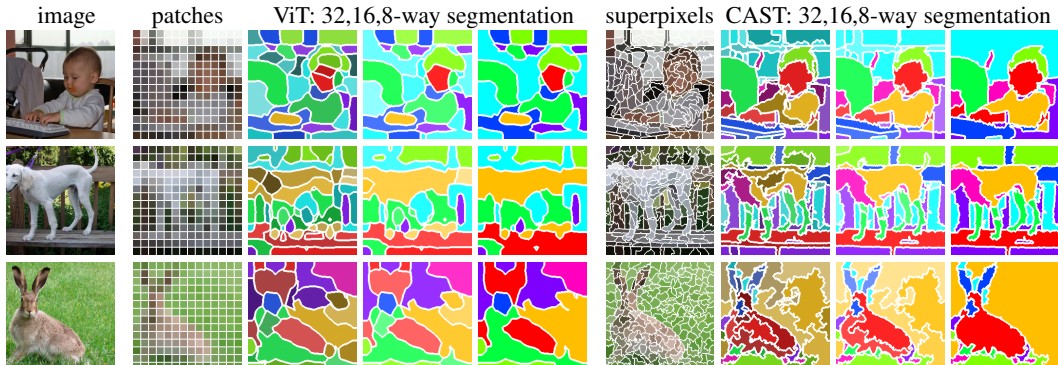

Figure 5: **CAST uncovers objects with complex contours, due to the use of not only superpixels but also progressive token pooling.** We train ViT and CAST on *unlabeled* ImageNet data using the MoCo objective (He et al., 2020). For the ImageNet image in Column 1 of each row, Columns 2-9 show respectively its square patches used by ViT, 32,16,8-way segmentations *derived* from ViT tokens via *fine-to-coarse* K-means clustering, superpixels used by CAST, and 32,16,8-way segmentations *generated* by CAST. Our color scheme has *coarse-to-fine* consistency: Colors in 8-way segmentations are matched between ViT and CAST, while colors in 16(32)-way segmentations have the same hues as 8-way but vary in saturation(value) to reflect finer details. Our results more closely follow visual contours and successfully uncover entire objects with details like *neck*, *thin legs*, and *long ears*.

Our model also offers flexibility in segmentation granularity *during inference*, unlike GroupViT (Xu et al., 2022) and HSG (Ke et al., 2022), which rely on a fixed number of learnable queries for next-level segments that cannot be altered post-training. In our approach, the number of clustering centroids, which can differ from the training configuration, determines segmentation granularity. Specifically, we employ the Farthest Point Sampling (FPS) (Qi et al., 2017) to select a subset of token features as initial centroids. This strategy ensures maximal coverage of the feature space without bias towards dominant features, leading to more robust segmentation (Appx. F).

**Architecture and Training**. CAST can be integrated into any existing ViT architecture by replacing its patch encoder with our segment encoder and by inserting our graph pooling module within the ViT blocks. We use SEEDS to extract superpixels and apply convolutional layers from (Xiao et al., 2021) to obtain initial segment features, ensuring a fair comparison with ViT. Following the original ViT architecture, we name our models CAST-(S/B), corresponding to ViT-(S/B). Segmentation granularity is set to $64, 32, 16, 8$ after $3, 3, 3, 2$ encoder blocks, respectively. The segments are referred to as level-$1, 2, 3, 4$ segments, starting with 196 superpixels at level-0. CAST can be trained by supervised learning as in DeiT (Touvron et al., 2021) and Segmenter (Strudel et al., 2021), or by self-supervised learning as in MoCo (He et al., 2020) on ImageNet (Deng et al., 2009) and COCO (Lin et al., 2014).

# 4 EXPERIMENTS ON HIERARCHICAL/FLAT SEGMENTATION & RECOGNITION

Our model has an internal hierarchical segmenter for recognition and can be trained solely using image recognition objectives. We study its performance and benefits in three tasks: **1)** unsupervised hierarchical segmentation and part-whole discovery, **2)** flat semantic segmentation, **3)** recognition. Additional ablation, results, experimental details, and visualizations can be found in the Appendix.

## 4.1 UNSUPERVISED HIERARCHICAL SEGMENTATION AND PART-WHOLE DISCOVERY

We consider three interesting baselines for CAST: ViT (Dosovitskiy et al., 2020), HSG (Ke et al., 2022), and SAM (Kirillov et al., 2023). **1)** CAST, ViT, and HSG can all be trained on the same *unlabeled* data. In contrast, SAM has been pre-trained on large-scale, pixel-wise labeled images and can be used directly. As a boundary segmenter, SAM only knows *where* but *not what* segments are in the image. **2)** ViT does not produce a hierarchical segmentation. We construct a hierarchy by applying fine-to-coarse K-means clustering of its final-layer tokens, trained without multiscale parsing and shown to be effective for detection (Li et al., 2022c) and segmentation (Kirillov et al., 2023). **3)** Both HSG and CAST produce a consistent hierarchical segmentation based on superpixels. However, HSG is designed *for segmentation*, whereas CAST is designed *for recognition*. As HSG uses ResNet50, we compare HSG with similarly sized CAST-S. **4)** SAM outputs segmentations at multiple granularities, but it does not enforce consistency among them. We report on both SAM-H, the largest model, and SAM-B, the smallest model which matches ViT-B and CAST-B in size.

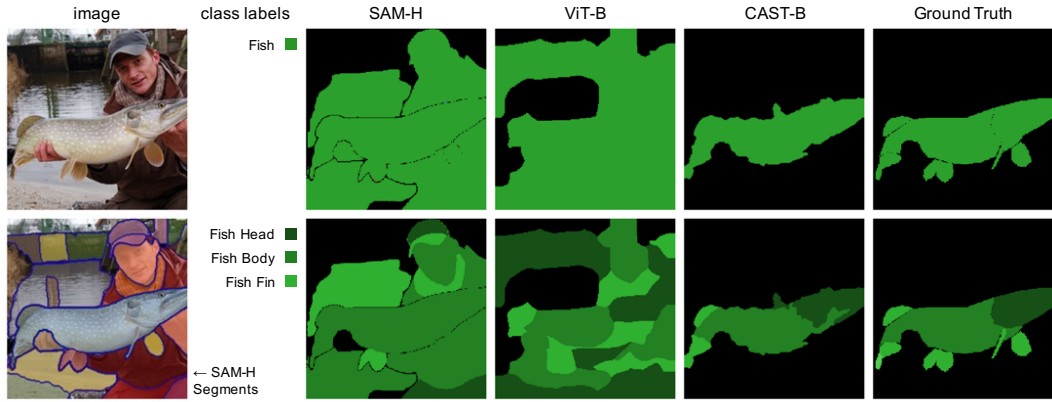

Figure 6: **CAST enhances recognition with a hierarchical segmentation that groups parts into wholes without supervision, whereas both ViT and SAM perform poorly: They lack superpixels to capture visual details and a hierarchical segmenter to simplify recognition.** SAM is pretrained on 11M images and 1B masks, while ViT and CAST are self-supervised on 1M ImageNet images, all producing segmentations on PartImageNet without semantics. We name each segment using OVSeg (Liang et al., 2023) for evaluation. For SAM-H output in Column 1 Row 2, we first name segments in descending size using the *object* vocabulary and then name *parts* based on identified objects. For ViT, we derive a hierarchical segmentation by applying fine-to-coarse K-means clustering of its last-layer tokens. For ViT and CAST, we use the object vocabulary to name 8-way segmentations and the part vocabulary for 16-way segmentations. Here the *fish* has distinct *head* and *fin* continuing onto the *body*, and its contrast with the background varies along the *body*. SAM struggles to parse *the fish* into parts, while ViT fails to separate them from the background, misleading recognition. In contrast, our *unsupervised* CAST accurately uncovers the whole *fish* with its distinct parts: *head*, *body*, and *fin*.

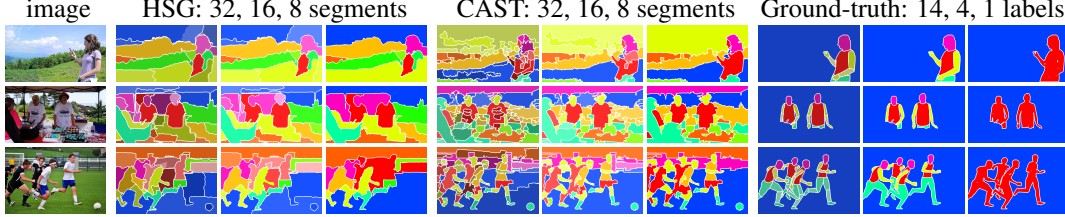

Figure 7: **Unsupervised hierarchical segmentation aligns more closely with semantics in CAST than in HSG.** We compare 32, 16, 8-way segmentation against fine, coarse, and whole person parsing respectively. To visualize a consistent hierarchy in color, we first match the 8-way segmentation to the 1-label ground truth in hue, then vary saturation for 16-way and value for 32-way. Overall, CAST is better than HSG on DensePose at separating human bodies from their backgrounds at all levels.

Fig. 5 has shown that CAST outperforms ViT at uncovering objects with complex contours and fine details. We now compare unsupervised CAST to SAM and ViT on PartImageNet for deformable and rigid object parsing (Fig. 6), and to HSG on DensePose for human body parsing (Fig. 7).

**PartImageNet** (He et al., 2022) is a subset of ImageNet additionally annotated with 11 objects (e.g., *Biped*) and 40 parts (e.g., *Biped Head*) (details in Appx. B.1). We benchmark SAM (*as is*), ViT and CAST (*trained self-supervisedly on ImageNet*) on PartImageNet. Since they all produce segmentations without class labels, for the sake of evaluation, we name each segment using an open-vocabulary segment classifier, OVSeg (Liang et al., 2023). For SAM, we use the default setting to produce a series of binary masks that may overlap. Since larger segments are more likely whole objects and smaller segments parts, we name segments in one pass using the PartImageNet *object* vocabulary. We name each mask starting with the largest, adjusting the smaller ones to remove any overlap with it. In a second pass, we name each identified segment using the PartImageNet *part* vocabulary. For ViT and CAST, we first name objects in a coarse (8-way) segmentation and then name parts within each object in a fine (16-way) segmentation. See Appx. E.3 for more details.

Fig. 6 shows that unsupervised CAST groups visual details into larger wholes that separate objects from the background, enabling accurate segment naming. Neither SAM nor ViT maintains segmentation consistency across granularities during training, resulting in coarse and fine segmentations that are error-prone and *compete* rather than *simplify* for recognition at the end.

Table 1: **Unsupervised CAST surpasses SAM and ViT in accuracy and efficiency for segmenting object parts and wholes on PartImageNet.** We measure region mIoU and boundary F-score, at first object and then part levels. SAM, pretrained on large-scale high-resolution images with pixel-wise labels, excels at delineating boundaries: At the object(part) level, SAM is 10(5)% over CAST, CAST is 11(1)% over ViT. However, on region accuracy, CAST outperforms SAM (ViT) by 8(4)% for objects and 1(1)% for parts, with only 4(72)% of their GFLOPS.

Table 2: **CAST surpasses HSG in unsupervised hierarchical semantic segmentation on DensePose, despite its focus on recognition instead of segmentation.** Training CAST-S and HSG on *unlabeled* COCO data, we evaluate 32,16,8-way segmentations on ground-truth 14,4,1-label human body parsing respectively on DensePose. We measure precision (P), recall (R), and F-score (F) on between segmentation and ground-truth human body. CAST consistently excels, especially in whole body recall.

| model | GFLOPS | region mIoU | | boundary F-score | |
|---|---|---|---|---|---|
| SAM-B | 677 | 18.03 | 10.15 | 20.71 | 7.25 |
| SAM-H | 3166 | 21.97 | 12.07 | **32.66** | **11.82** |
| ViT-B | 18 | 25.34 | 11.74 | 10.92 | 4.64 |
| CAST-B | **13** | **29.66** | **13.20** | 22.32 | 6.52 |

| P R F | 14 labels | | | 4 labels | | | 1 label | | |
|---|---|---|---|---|---|---|---|---|---|
| HSG | 20.7 | 18.6 | 19.6 | 24.1 | 30.6 | 26.9 | 20.5 | 36.1 | 26.2 |
| CAST | **21.1** | **24.1** | **22.5** | **24.8** | **33.2** | **28.4** | **26.3** | **44.9** | **33.2** |
| gain | 0.4 | 5.5 | 2.9 | 0.7 | 2.6 | 1.5 | 5.8 | 8.8 | 7.0 |

Table 1 shows that our *unsupervisedly* trained CAST outperforms not only ViT by 4% (in mIoU), but also SAM by 8% on object segmentation, which is *supervisedly* trained on SA-1B and $243\times$ larger in GLFOPS! While SAM can produce a high-quality segmentation, it does not organize these segments and simplify them for recognition, especially as PartImageNet only annotates a single prominent object in an image. The *human* segment in Fig. 6, with boundaries shaped as the occluding *fish*, is mistaken by OVSeg as Category *fish* instead of *biped*. Such interference from border-ownerships has long been acknowledged in human perception research, exemplified by *Bregman's scrambled B illusion* (Bregman, 1981). In addition, CAST is also significantly more efficient. SAM requires **1)** an additional mask decoder to predict pixel-segment assignments in a high-dimensional feature space, and **2)** multiple inferences guided by various point prompts. CAST produces a hierarchy of segments starting from color features in a single inference pass, with only 4% of SAM-H's GFLOPS!

**DensePose** (Alp Güler et al., 2018) has complex multi-person scenes, each individual annotated with 14 body parts. We group these labels into 4 categories: (*head*, *torso*, *upper limb*, and *lower limb*. We train HSG and CAST-S self-supervisedly on COCO (Lin et al., 2014) and compare their 32, 16, 8-way segmentations on DensePose against fine (14 labels), coarse (4 labels), and whole person (1 label) parsing respectively (Fig 7). Table 2 shows that CAST surpasses HSG in both precision and recall at every level. While HSG optimizes feature distinction across segments at all levels, CAST concentrates this optimization at the final image level only. Therefore, approaching hierarchical segmentation not as a standalone goal but as integral to recognition in fact enhances segmentation!

## 4.2 SEMANTIC SEGMENTATION

Now we can learn image segmentation without pixel-wise labeling, but with only an image-level recognition objective. We compare CAST with ViT on utilizing such a segmenter for downstream semantic segmentation tasks (Table 3). **1)** We first train both models on *unlabeled* COCO data, and then evaluate their segmentation *before* and *after* fine-tuning on PASCAL VOC (Everingham et al., 2010). **2)** We first train both models on *labeled* ImageNet data, then fine-tune with Segmenter (Strudel et al., 2021) and test on ADE20K (Zhou et al., 2019) and PASCAL Context (Mottaghi et al., 2014).

Table 3: **CAST outperforms ViT with superpixels and token pooling on flat semantic segmentation by unsupervised or supervised learning.** We report both region mIoU and boundary F-score to benchmark CAST-S and ViT-S: **a)** on PASCAL-VOC after training on *unlabeled* COCO, and **b)** on ADE20K and PASCAL-Context following training on *labeled* ImageNet with subsequent tuning. We explore ViT variants with superpixels replacing traditional patches and graph pooling for token reduction. CAST consistently outperforms, validating the efficacy of both modifications.

| **a)** *self-supervised* on COCO | | | | | | **b)** *supervised* on ImageNet, tuned for | | | | |
|---|---|---|---|---|---|---|---|---|---|---|
| test on PASCAL-VOC | | before tuning | | after tuning | | | | ADE20K | | P-Context |
| ViT-S | | 30.9 | 16.1 | 65.8 | 40.7 | ViT-S | 41.7 | 33.9 | 48.3 | 42.0 |
| ViT-S but with superpixels | | 32.2 | 21.2 | 66.5 | 46.7 | - | - | - | - | - |
| ViT-S but with token pooling | | 34.5 | 19.8 | 67.2 | 41.9 | - | - | - | - | - |
| CAST-S | | **38.4** | **27.0** | **67.6** | **48.1** | CAST-S | **43.1** | **36.5** | **49.1** | **44.1** |

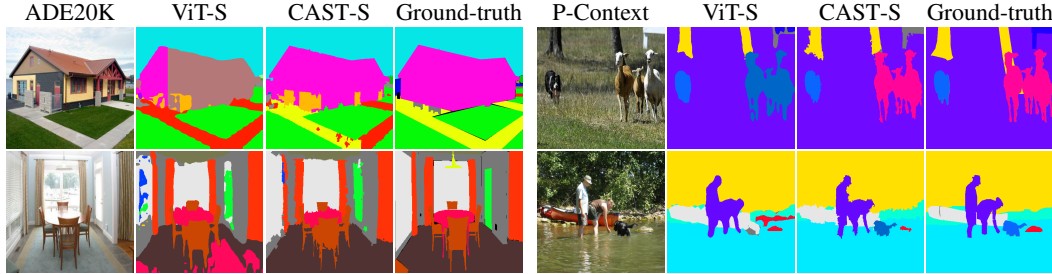

Figure 8: **CAST pretrained for recognition generalizes better than ViT to semantic segmentation.** Both ViT-S and CAST-S are pre-trained on labeled ImageNet and fine-tuned for semantic segmentation on ADE20K or PASCAL-Context. Unlike ViT, CAST can not only recognize and segment *large wholes* (*house*, *floors*), but also closely adhere to visual contours of small ones (*dog*).

We follow SegSort (Hwang et al., 2019) on segment retrieval for unsupervised evaluation, and fine-tune the model alongside a pixel-wise softmax classifier for supervised evaluation.

Table 3 shows that CAST consistently outperforms ViT, and both our components, superpixels and token pooling, contribute to performance gains, most notably increasing 8% in region mIoU and 11% in boundary F-score before fine-tuning. Fig. 8 shows that CAST retains these benefits even after supervised training and fine tuning, recognizing both wholes and details more accurately.

## 4.3 RECOGNITION AND REFINEMENT OF SEGMENTATION THROUGH FEEDBACK

We now study CAST's performance on recognition and its feedback connection to segmentation.

**Classification.** We compare CAST-S to ViT-S and Swin-T from Swin Transformer (Liu et al., 2021), both designed solely for recognition and with similar GFLOPS. All three models are self-supervised on IN-1K and tested on IN-1K and IN-100 using linear probing. Table 4 shows that, with hierarchical token pooling, CAST outperforms ViT and Swin with 30% fewer GFLOPS. Notably, only CAST can directly output a hierarchical segmentation, with a stronger object-centric attention (Appx. A.5).

| Model | GFLOPS | IN-100 | IN-1K |
|---|---|---|---|
| ViT-S | 4.7 | 78.1 | 67.9 |
| Swin-T | 4.5 | 78.3 | 63.0 |
| CAST-S | **3.4** | **79.9** | **68.1** |

Table 4: **CAST outperforms ViT and Swin on ImageNet classification with higher efficiency.** We report the top-1 linear probing accuracy of ViT, Swin, and CAST, self-supervised on IN-1K and tested on IN-100/1K. CAST achieves higher accuracies using about 30% fewer GFLOPS.

**Test-time adaptation.** CAST not only learns an internal segmenter in a feed-forward hierarchy towards recognition, but can also revise the segmenter in a feedback reverse hierarchy, concurrently enhancing both segmentation and recognition (Fig. 3). Fig. 9 shows that ambiguous recognition can be solidified by maximizing the winning activation, modifying segmentation and in turn enhancing both segmentation and recognition concurrently. See more details in Appx. C.

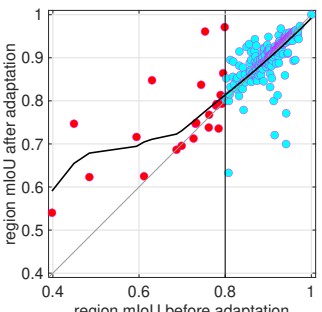

Figure 9: **CAST can be adapted during test time to solidify ambiguous recognition and refine segmentation.** For the binary *dog* classification task on PartImageNet shown in Fig. 3, We assess region mIoU on PartImageNet for CAST's 8-way segmentation against the ground-truth *dog* mask *before* and *after* adaptation (Fig. 3). When the mIoU is initially lower than threshold 0.8 (vertical line), it has a large and positive gain mostly (red dots above the diagonal line). The average gain (black line) is *large* (*zero*) with poor initial segmentation when adaptation is *most* (*not*) needed. Our concurrent segmentation and recognition enhances each other in this adaptation process.

**Summary.** We propose to include segmentation as part **of** recognition, to learn it **by** image recognition objectives, and to infer it concurrently with and **for** recognition. We develop our model, CAST, by utilizing superpixels and token pooling. It uncoveres parts and wholes, surpassing SAM, HSG, and ViT in both accuracy and efficiency on various segmentation and recognition tasks.

## ACKNOWLEDGEMENTS

This work was supported, in part, by US NSF 2131111, NSF 2215542, NSF 2313151, and Bosch gift funds to S. Yu at UC Berkeley and the University of Michigan. We thank Jyh-Jing Hwang for earlier insightful discussions that led to this work.

## ETHICS STATEMENT

Our CAST provides a fine-grained understanding of part-to-whole structures, unlocking new possibilities while also carrying the potential for misuse. For instance, detailed structures like human poses could be used to enhance the quality of DeepFake videos (Masood et al., 2023). However, it's crucial to emphasize that technology itself is not the issue; the responsibility falls on those who wield it. We aspire for our CAST to bring about more benefits than these potential risks.

## REPRODUCIBILITY STATEMENT

We provide method and experimental details in Appx. E and Appx. F, and our code on GitHub.

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

APPENDIX

# A  ABLATION STUDIES AND ANALYSES

## A.1  ABLATION STUDY ON DESIGN CHOICES

Table 5 presents ablation studies on design choices in our framework. **(a)** We explore various token pooling methods, with our GraphPool module excelling. Notably, FINCH (Sarfraz et al., 2019) lags in performance, showcasing the challenge of adaptive pooling. **(b)** We examine different centroid initialization methods, with Farthest Point Sampling (FPS) significantly outperforming others. FPS not only samples informative tokens but also enhances discriminative token selection, preserving fine-grained visual information. **(c)** We investigate optimal segment granularities, fixed for both training and inference, to achieve a balance between model efficiency and task performance. **(d)** We showcase that our model can adapt to varying segment granularities during inference.

Table 5: We compare the design choices for (a) token pooling, (b) cluster centroid initialization, (c) token granularities applied to both training and testing, and (d) token granularities of a trained model varied during testing. We report the linear probing accuracy of MoCo-trained CAST-S on IN-100. † indicates that the methods have been re-implemented.

| (a) Pooling | Acc. |
|---|---|
| Graph Pooling | **79.9** |
| Random Sampling | 55.8 |
| K-Means | 73.9 |
| K-Medoids | 72.3 |
| FINCH† (Sarfraz et al., 2019) | 63.3 |
| Token Pooling† (Marin et al., 2021) | 75.8 |
| CTM† (Zeng et al., 2022) | 72.2 |
| ToMe† (Bolya et al., 2023) | 78.1 |

| (b) Centroids initialization | Acc. |
|---|---|
| Farthest Point Sampling | **79.9** |
| Random Sampling | 71.2 |
| PoWER-BERT (Goyal et al., 2020) | 71.6 |

| (c) Token granularity (Train=Test) | GFLOPS | Acc. |
|---|---|---|
| $196, 32, 16, 8$ | 3.0 | 78.8 |
| $196, 64, 32, 16$ | 3.4 | **79.9** |
| $196, 128, 64, 32$ | 4.3 | **79.9** |
| $324, 64, 32, 16$ | 4.6 | 79.8 |
| $324, 128, 64, 32$ | 5.4 | 80.4 |

| (d) Token granularity (Train≠Test) | GFLOPS | Acc. |
|---|---|---|
| Train: $196, 64, 32, 16$ | 3.4 | **79.9** |
| Test: $196, 32, 16, 8$ | 3.0 | 79.2 |
| Test: $196, 128, 64, 32$ | 4.3 | 79.5 |
| Test: $324, 64, 32, 16$ | 4.6 | 78.8 |
| Test: $324, 128, 64, 32$ | 5.4 | 79.3 |

A.2   ABLATION STUDY ON SUPERPIXEL METHODS

We test the robustness of CAST using superpixels generated by different algorithms. In particular, we compare SEEDS (Bergh et al., 2012) against SLIC (Achanta et al., 2012) on classification and semantic segmentation. We train CAST with self-supervised learning on IN-1K and COCO for classification and segmentation. As shown in Table 6, we report the linear probing accuracy on ImageNet-1K and mIoU before (left) and after (right) fine-tuning on VOC. Our method is robust to different choices of superpixel algorithms, while, SEEDS achieves better performance than SLIC.

Table 6: SEEDS outperforms SLIC on classification and semantic segmentation. We report top-1 accuracy and mIoU for classification on IN-1K and segmentation before / after fine-tuning on VOC. Our CAST achieves better performance using superpixels generated by SEEDS.

|  | Classification | Segmentation |
|---|---|---|
| SEEDS | 68.1 | 38.4/67.6 |
| SLIC | 65.6 | 37.7/65.7 |

Figure 10 illustrates the superpixels obtained using SEEDS and SLIC. SEEDS superpixels exhibit superior alignment with object boundaries, particularly for small objects, compared to SLIC. This improvement stems from SEEDS being a boundary refinement algorithm, iteratively updating superpixels to better capture edges. In contrast, SLIC relies on K-Means clustering based on colors and connected-component constraints, which may be less effective in capturing the boundaries. Consequently, we can conclude that SEEDS superpixels, with their superior shape information, produce more compelling hierarchical segmentation for CAST.

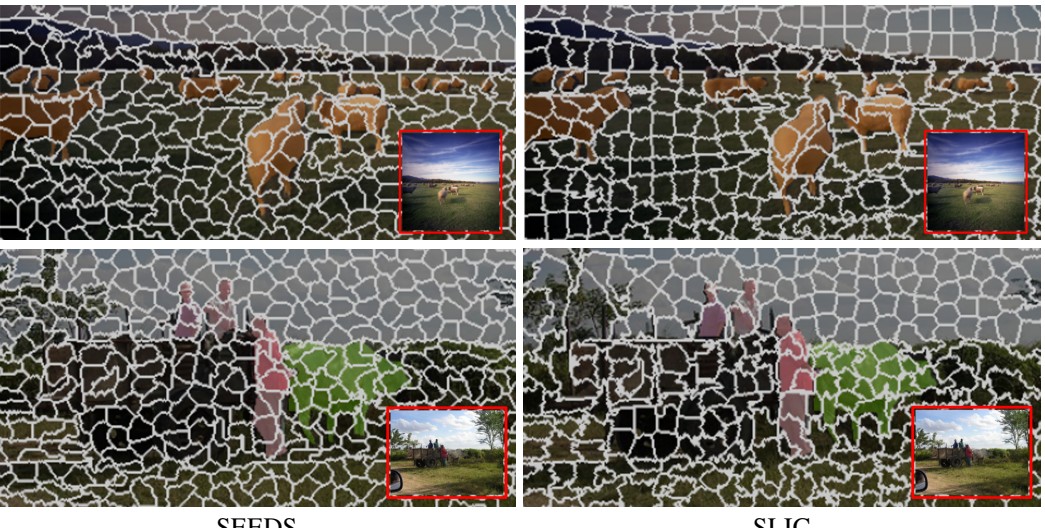

SEEDS                                              SLIC

Figure 10: From left to right, we show the superpixels (white contours) generated by SEEDS and SLIC algorithm, overlaid on the the original images along with the ground-truth semantic segmentation (colored regions). Superpixels generated by SEEDS are more precisely aligned with object boundaries, especially for small objects, than those generated by SLIC. Learning to generate superpixels along with feature learning would enable more precise superpixel partitioning and improve segmentation.

### A.3 LIMITATIONS OF SUPERPIXELS AND FAILURE CASES

Our method underperforms ViT baselines on Cityscapes dataset. The results show the limitation of using off-the-shelf superpixel algorithms. ViT-S achieves 74.6%, whereas, CAST-S and CAST-SD achieve 72.1% and 74.2% on the benchmark. As shown in Fig. 11, we observe that existing superpixel methods cannot pick out thin structures such as light poles and steel pipes from the scene. One possible reason is that such methods are only developed and tested on object-centric dataset (BSDS). Replacing these superpixel algorithms with learnable approaches could help to address such an issue.

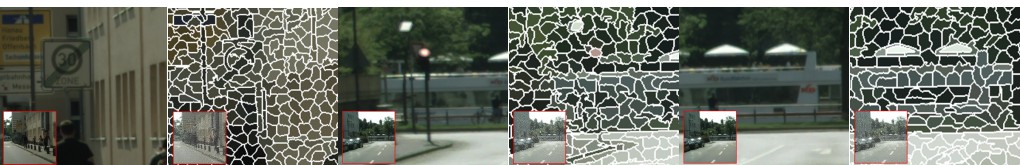

Figure 11: From left to right, we show the image and corresponding superpixels (white contours). Street scenes have many thin structures such as light poles and steel pipes, which superpixel algorithms often fail to pick out. Learning to generate superpixels could help solving such issues.

### A.4 ABLATION STUDY ON INFERENCE LATENCY

The computational cost of superpixel generation and graph pooling can be reduced by the decreased number of tokens required for computing self-attention. This advantage becomes more pronounced when using larger models, where self-attention blocks dominate the entire cost.

To validate this, we analyze the latency of model inference and superpixel generation. Our system comprises a 32GB Nvidia Titan V GPU card and two Intel(R) Xeon(R) CPU E5-2630 v4 processors, totaling 20 CPU cores. We utilize the PyTorch machine learning framework with 24 workers, a batch size of 64, and an image resolution set to 224x224.

In our system, CAST-B achieves a lower average inference latency of 217 ms compared to ViT-B with 273 ms. SEEDS takes 73 ms to generate superpixels from the batch of images. However, we remark that the current SEEDS implementation is not fully optimized. Employing GPU implementation or parallelizing the process with more CPU cores can alleviate the bottleneck in superpixel generation. Furthermore, the cost of superpixel generation becomes less significant with larger models, which are commonly used in practice.

In addition, we demonstrate that the cost of self-attention and graph pooling decreases as the number of tokens is reduced. Table 7 presents the computational cost of our graph pooling across layers, showing a significant reduction in cost as the number of tokens per layer decreases.

Table 7: FPS in our graph pool module requires additional computation. We report the inference time (ms) of each module with 384 channel dimensions and 256 batch sizes on IN-100. Optimizing the token sampling technique to increase model efficiency is our future work.

| #. of Tokens | Encoder Blocks | GraphPool (FPS) |
|---|---|---|
| 196 | 86.43 | 63.02 (37.64) |
| 64 | 25.4 | 18.2 (9.7) |
| 32 | 12.9 | 9.6 (3.0) |
| 16 | 5 | 6.1 (1.5) |

## A.5   VISUALIZATION OF ATTENTION MAPS

CAST provides more object-centric attention than ViT. We compare models self-supervised on IN-1K and evaluate them on Pascal VOC. We follow Joulin et al. (2010) to generate figure-ground segmentation from multi-head attention and measure mIoU between true segmentations. Fig. 12 visualizes an example of an attention map and the mIoU of each model.

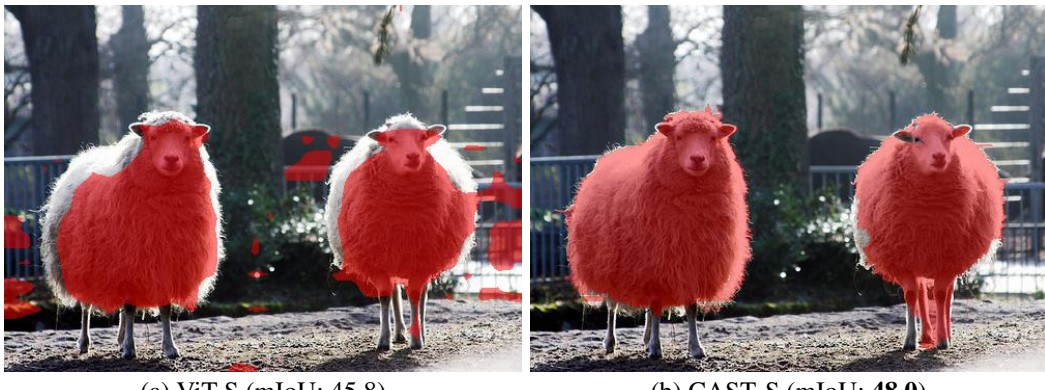

(a) ViT-S (mIoU: 45.8)          (b) CAST-S (mIoU: **48.0**)

Figure 12: CAST attention is more object-centric than ViT. We visualize figure-grounded attention maps of ViT and CAST. The bracket denotes the mIoU between attention and true segmentations. The CAST attention map covers the entire sheep, while ViT only captures a part of them.

## B    ADDITIONAL PART-TO-WHOLE RESULTS

### B.1    EXAMPLES FOR PARTIMAGENET

We present the annotation lists and visual examples of PartImageNet. It contains 11 object classes from 2 categories (animals vs. things), and each object class contains multiple part classes, resulting in a total of 40 parts. Part segments are constrained to be subregions of object segments.

Table 8: PartImageNet contains 11 objects and 40 parts annotations. Each object segment is partitioned into part segments. For example, a quadruped segment consists of quadruped head, quadruped body, quadruped foot, and quadruped tail segments. In our experiments, we first predict the object labels, and then predict part labels conditioned on the object to enforce consistent prediction; the subregion of a quadruped should be one of head, body, foot, or tail.

| Category | Object | Part |
|---|---|---|
| Animal | Quadruped | Head, Body, Foot, Tail |
|  | Biped | Head, Body, Hand, Foot, Tail |
|  | Fish | Head, Body, Fin, Tail |
|  | Bird | Head, Body, Wing, Foot, Tail |
|  | Snake | Head, Body |
|  | Reptile | Head, Body, Foot, Tail |
| Things | Car | Body, Tire, Side Mirror |
|  | Bicycle | Head, Body, Seat, Tire |
|  | Boat | Body, Sail |
|  | Aeroplane | Head, Body, Wing, Engine, Tail |
|  | Bottle | Body, Mouth |

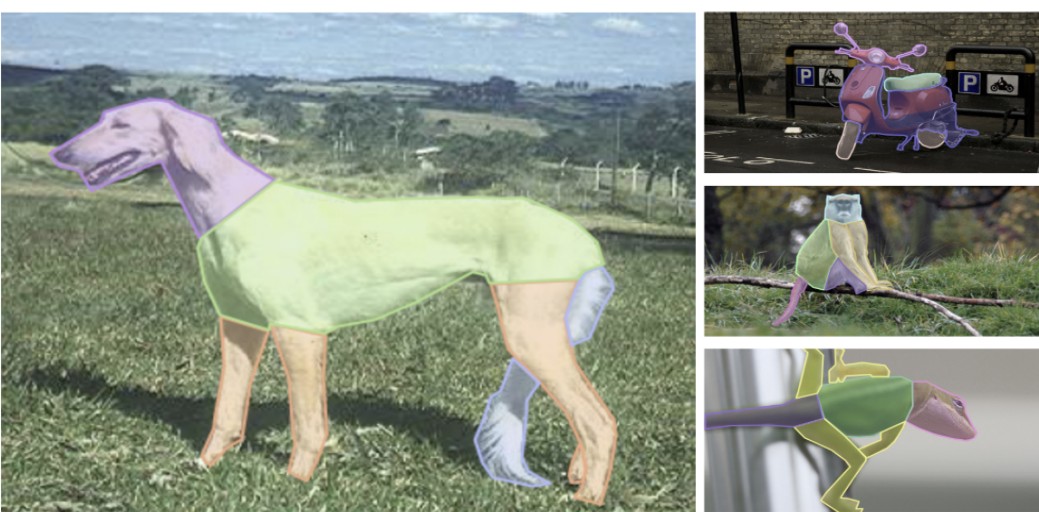

Figure 13: Example image from PartImageNet. Copied Figure 1 from the original paper (He et al., 2022). The quadruped (dog) segment consists of head, body, foot, and tail part segments.

## B.2 CLASS-WISE PERFORMANCE

We present the class-wise performance of part-to-whole recognition on PartImageNet in Table 9. CAST outperforms SAM and ViT in most classes, except for reptile and bottle. To further investigate this, we illustrate visual examples where 1) CAST performs well (quadruped), 2) ViT performs well (reptile), and 3) SAM performs well (bottle) in Fig. 14.

Table 9: CAST outperforms SAM and ViT in most classes. We report the region mIoU on object annotations of SAM, ViT, and CAST for each class in PartImageNet, as well as the average across classes in each Animal or Things category. Bold denotes the best value among the methods. The Aeroplane class shows '-' for all methods since it is not contained in the validation set.

| | Animal | | | | | | |
| | Quadruped | Biped | Fish | Bird | Snake | Reptile | Avg. |
|---|---|---|---|---|---|---|---|
| SAM-B | 18.18 | 14.98 | 16.09 | 24.55 | 19.87 | 14.78 | 18.08 |
| SAM-H | 29.00 | 17.25 | 19.97 | 21.35 | 20.57 | 13.05 | 20.20 |
| ViT-B | 29.61 | 15.17 | 24.34 | 30.02 | 22.97 | **20.60** | 23.79 |
| CAST-B | **37.47** | **20.94** | **29.32** | **36.97** | **23.72** | 20.34 | **28.13** |

| | Things | | | | |
| | Car | Bicycle | Boat | Aeroplane | Bottle | Avg. |
|---|---|---|---|---|---|---|
| SAM-B | 15.89 | 15.16 | 12.14 | - | 31.30 | 18.62 |
| SAM-H | 29.09 | 14.20 | 9.34 | - | **35.50** | 22.03 |
| ViT-B | 32.15 | 18.33 | 20.19 | - | 31.59 | 25.57 |
| CAST-B | **36.39** | **20.26** | **22.61** | - | 27.36 | **26.66** |

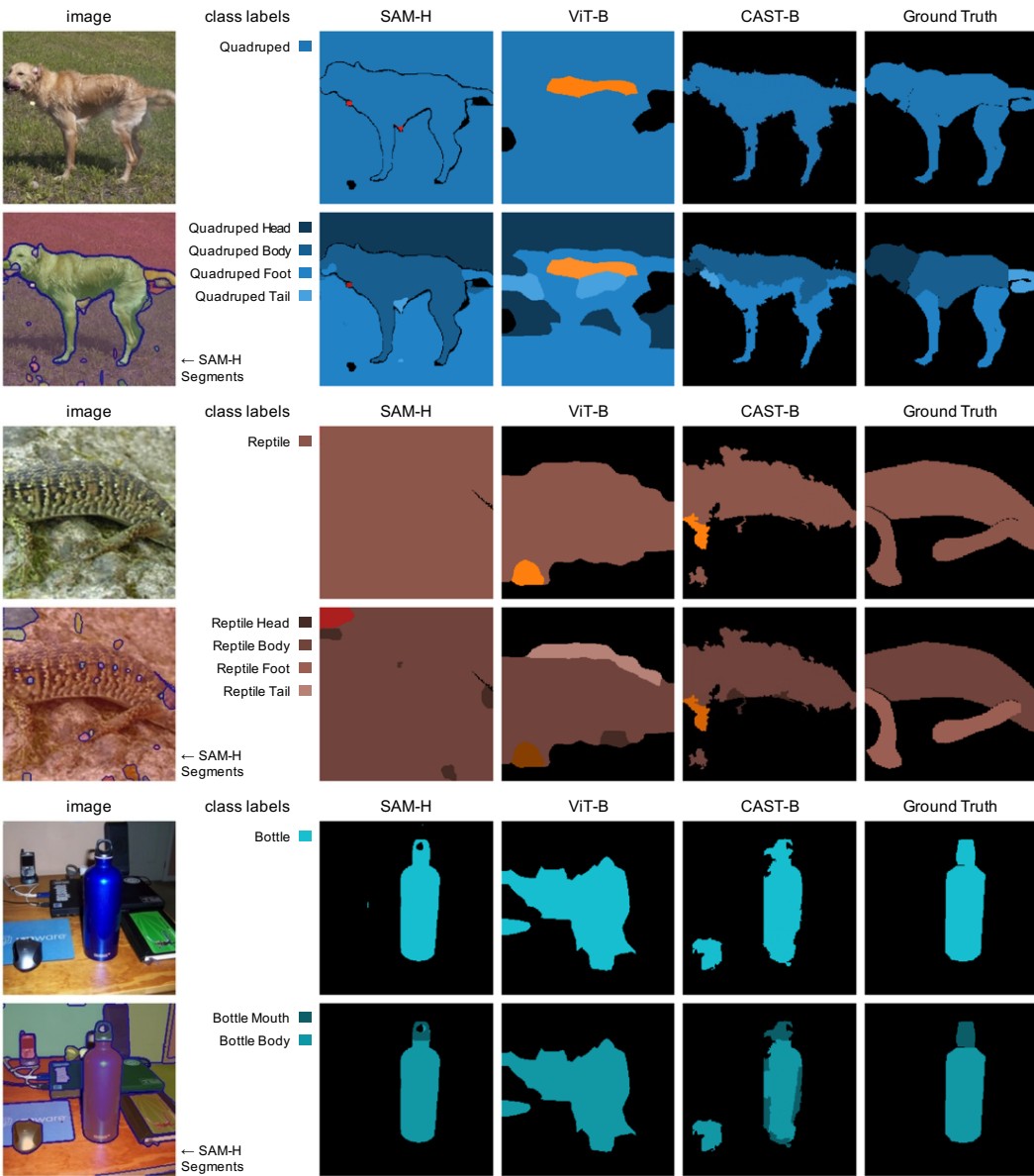

Figure 14: Visual examples of object and part segmentations on PartImageNet using SAM, ViT, and CAST. We demonstrate three cases where each model excels. 1) CAST excels for quadrupeds. Here, CAST succeeds in capturing the parts properly, such as the head, body, and feet. SAM could capture the dog, but fails to partition the parts such as the head. 2) ViT excels for reptiles. SAM failed to capture the reptile due to camouflage. CAST captured the body but missed the feet. In contrast, ViT does not provide detailed boundaries; it covers the whole body of the reptile, giving a better region mIoU than others. 3) SAM excels for bottles. Here, SAM clearly describes the boundaries of the bottle. However, CAST mistakenly clusters the mouse with the bottle.

## C    EXTENSION TO TEST-TIME ADAPTATION

CAST only considers the top-down pathway during training, guided by the recognition objectives. This knowledge is encoded in the model and reflects the bottom-up pathway during inference. While this enables the model to learn the general top-down elements, the segmentations will not change based on what the top layers predict at inference time.

To extend CAST by incorporating the top-down pathway during inference, we employ test-time adaptation (TTA) (Sun et al., 2020), specifically TENT (Wang et al., 2021a), with the classifier trained on top of the CAST backbones. We apply TENT to each sample to adapt the model and maximize its prediction confidence. As a result, CAST refines its object segments to align with its initial belief: If this image depicts the predicted class, which parts contribute to this prediction?

### C.1    IMPLEMENTATION DETAILS

We use the CAST model pretrained on ImageNet with the MoCo objective as our base model. We trained a dog vs. non-dog classifier on PartImageNet training data and evaluate it on the validation split. The classifier is defined as the average of normalized embeddings of training data that belong to the class. In other words, we define a class vector as the center of training data for each class. To infer the class, we compute the cosine similarity between the test embedding and class vectors and apply a temperature of $0.07$ to the softmax classifier, using these cosine similarities as logits.

For each sample, we calculate the prediction entropy loss and update the model using a single SGD with lr=1.0. We only update the normalization layers while keeping other parameters frozen, following the practice of TTA. Note that no label or batch information is used in this process, and the model is updated solely based on the inference of a single instance.

We evaluate the evolution of segmentation before and after TTA. To this end, we compute the object segmentation mask by assigning a label to each segment. We define this segment label as the majority of the pixel labels, using the ground-truth segmentation masks. We chose this strategy for simplicity, but one can also predict the segment labels using OVSeg, as we discussed previously. We measure the average of mIoU for the foreground (dog) and background, treating them as two classes.

## C.2 VISUAL EXAMPLES BEFORE AND AFTER TTA

Fig. 15 illustrates how TTA improves CAST segmentation by more accurately capturing object shapes and minimizing attention on irrelevant details. This improvement is especially notable for challenging samples where the original CAST performs poorly. To validate this, we present both successful cases where the mIoU significantly increases before and after TTA (rows 1-3) and failure cases where the mIoU decreases (rows 4-5). In the failure cases, TTA mistakenly forces CAST to cluster parts of dogs with the background, solely focusing on the most distinct part of the dog.

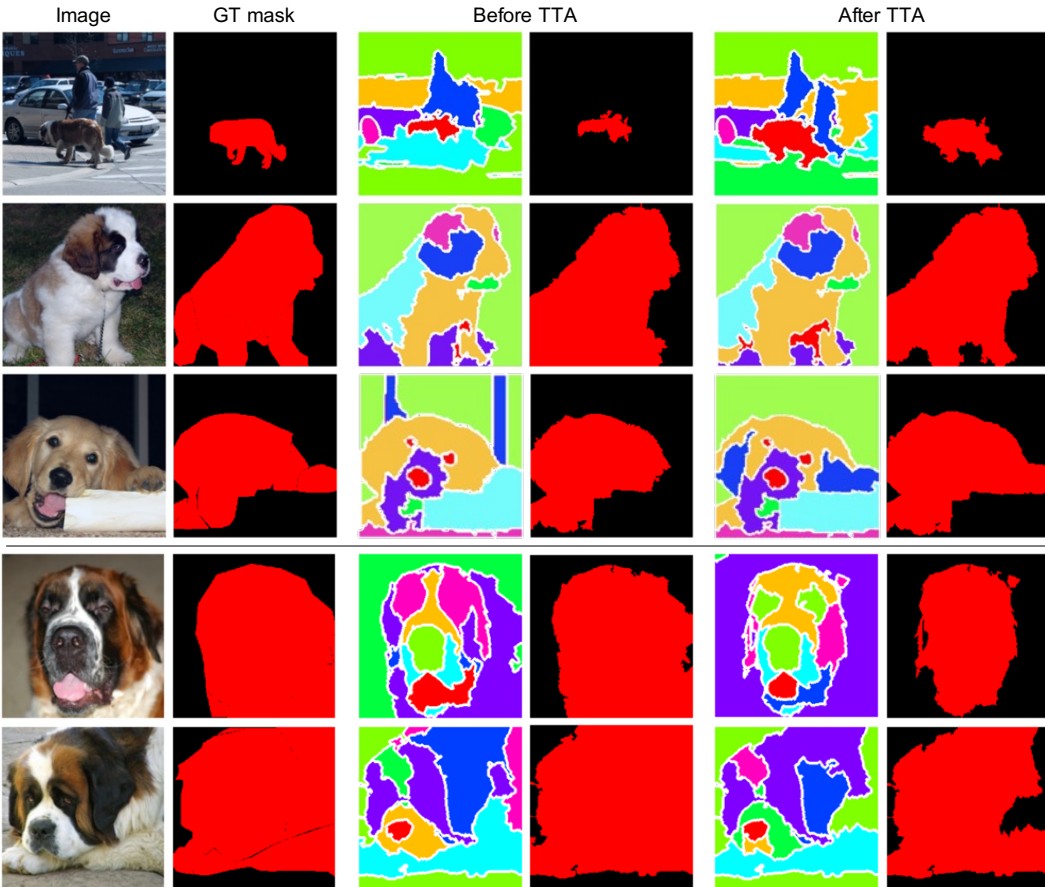

Figure 15: Visual examples of learned clusters (left) and predicted object segmentation for dogs (right) using CAST models before and after applying test-time-adaptation (TTA). The segmentation is given by assigning labels for each segment by majority vote using GT pixel labels. We showcase both successful cases (rows 1-3) and failure cases (rows 4-5). TTA enhances CAST segmentation by improving object shape capture, such as avoiding missed legs in rows 1-3, while also reducing attention to unnecessary details, like window frames in row 3. However, in the failure cases, CAST focuses solely on the distinct part and merges other parts with the background after TTA.

# D  ADDITIONAL RELATED WORKS

**Image segmentation** partitions an image into coherent regions. Classic methods consist of two steps: extracting local features and clustering based on criteria like mode-finding (Comaniciu & Meer, 2002; Banerjee et al., 2005) or graph partitioning (Felzenszwalb & Huttenlocher, 2004; Shi & Malik, 2000; Malik et al., 2001; Yu & Shi, 2003a; 2004). They often output hierarchical segmentation for human perception comparison (Arbelaez et al., 2010). To prevent object boundary ambiguities, common approaches rely on contour detection (Hwang & Liu, 2015; Xie & Tu, 2015), iteratively removing contours for multi-scale segmentations (Arbelaez et al., 2010). In contrast, our work operates directly on segments, providing proper boundaries without using contour proxies.

**Self-supervised segmentation and representation learning** aims to learn how to segment images, often with corresponding features, solely from images and without human-annotated segmentations. Recent works can be categorized into three camps: **1)** Leveraging self-supervised image recognition, models are transferred to segmentation by increasing location sensitivity (Wu et al., 2018; He et al., 2020; Chen et al., 2020; Wang et al., 2021c), implementing contrastive loss across views (Wang et al., 2021b), or applying stronger augmentation and constrained cropping (Selvaraju et al., 2021; Mo et al., 2021). **2)** Learning a pixel-wise cluster predictor maximizes mutual information between cluster predictions on augmented views of the same instance at corresponding pixels (Ji et al., 2019; Ouali et al., 2020). **3)** Learning a pixel-level feature encoder maximizes discrimination between pixels based on contour-induced segments (Hwang et al., 2019), pre-computed region hierarchies (Zhang & Maire, 2020), or feature-induced hierarchical groupings (Ke et al., 2022), deriving segmentation from pixel feature similarities. Our work unifies the first and third approaches, training CAST with a self-supervised image recognition framework but obtains hierarchical segmentation for free.

**Object-centric learning** aims to learn a structured representation of images that disentangles objects from the background (Eslami et al., 2016; Mo et al., 2019). Several prior works have attempted to incorporate this concept into the attention mechanism, such as combining it with vision transformers (Locatello et al., 2020; Xu et al., 2022; Kang et al., 2022; Shi et al., 2023). This approach not only discovers object segmentations but also improves robustness against distribution shifts. However, most prior works only focus on the object level. Our framework extends object-centric learning in a more fine-grained manner, discovering the continuum of part-to-whole relationships.

**Superpixels** are sets of locally connected pixels that encapsulate coherent structures, such as colors (Ren & Malik, 2003). They have found applications in various densely labeling tasks, including part parsing (Mori et al., 2004), saliency detection (Ren et al., 2013), and image segmentation (Gould et al., 2008; Fulkerson et al., 2009; Sharma et al., 2014; Gadde et al., 2016; Wei et al., 2018). More recently, Zhang et al. (2022) addressed semantic segmentation by replacing patches with superpixel tokens in ViT architectures. Our model takes a step further by constructing a segment hierarchy and simultaneously performing both segmentation and classification.

**Token pooling** aims to merge or prune tokens of (vision) transformers. Numerous methods have been proposed, but their primary goal was to improve efficiency, and they used square patches as usual (Sarfraz et al., 2019; Marin et al., 2021; Zeng et al., 2022). In contrast, our CAST employs superpixels as visual units, which provide a unique benefit of producing hierarchical segmentation for free. Additionally, our ablation study demonstrates that our proposed graph pooling outperforms the state-of-the-art token pooling methods like ToMe (Bolya et al., 2023) for our purpose.

# E  ADDITIONAL METHOD DETAILS

## E.1  ALGORITHM

We present the pseudo code of the Graph Pooling module and our CAST framework.

---

**Algorithm 1:** GraphPool

**Input:** Feature $Z$ and number of clusters $N$
**Output:** Coarse feature $Y$ and assignments $P$
Centroid indices $C \leftarrow \text{FPS}(Z, N)$
Refined feature $U \leftarrow \text{MSA}(Z) + Z$
Normalized feature
  $U \leftarrow U - \text{mean}(U) + \text{bias}$
Centroid feature $V \leftarrow \{U[c] | c \in C\}$
$P \leftarrow \text{softmax}(\kappa \frac{UV^\top}{\|U\|\|V\|})$
Project feature $Z' \leftarrow \text{MLP}(Z)$
Average pooled feature $Z' \leftarrow P^\top Z'./P^\top 1$
New centroid feature $Y \leftarrow \{Z[c] | c \in C\}$
Updated centroid feature $Y \leftarrow Y + Z'$

---

FPS: Farthest Point Sampling.
MSA: Multi-headed Self-Attention.
MLP: MultiLayer Perceptron
FC: Fully Connected Layer.
$\oplus$: Concatenation operator.

---

**Algorithm 2:** Overall framework

**Input:** Input image $I$, CNN features $X_{\text{cnn}}$, class token $X_{\text{class}}$, position encoding $E_{\text{pos}}$, # of segments $N_l$ at level $l$
**Output:** Feature $\mathbf{f}_{\text{class}}$ or $\mathbf{f}_{\text{seg}}$
Input segmentation $S_0 \leftarrow \text{Superpixel}(I, N_0)$
Input tokens $X_s \leftarrow \{\frac{1}{|a|} \sum_{i \in a} X_{\text{cnn}}[i] | a \in S_0\}$
**for** $l = 0 \ldots L$ **do**
  **if** $l = 0$ **then**
    Initial segment features
      $Z_0 \leftarrow [X_s + E_{\text{pos}}; X_{\text{class}}]$
  **else**
    Coarsened segment features and
      clustering assignments
      $Z_l, P_l \leftarrow \text{GraphPool}(Z_{l-1}, N_l)$
    Coarsened segmentation
      $S_l \leftarrow S_{l-1} \times P_l$
  **end**
  $Z_l \leftarrow \text{ViT\_Encoders}(Z_l)$
**end**
Class token $\mathbf{f}_{\text{class}} \leftarrow Z_L[0]$
Multi-level segment tokens $\mathbf{f}_{\text{seg}} \leftarrow \text{FC}(\oplus(Z_0[1 : N_0], \ldots, \text{Unpool}(Z_L[1 : N_L])))$

---

To obtain multi-level segment tokens, we unpool coarse tokens to the corresponding fine segments, concatenate the unpooled features with the fine tokens. The segment tokens $Z_l$ are scattered to the affiliated fine segments:

$$Z_{l-1}^{\text{unpool}}[a] = Z_l[c], c = \text{argmax}_{c \in C} P_l[a, c]. \tag{4}$$

## E.2 COMPARISON WITH VIT

We highlight the technical differences between our CAST and prior works, including CNN, ViT, and Token Pooling in Fig. 16 and in Fig. 17.

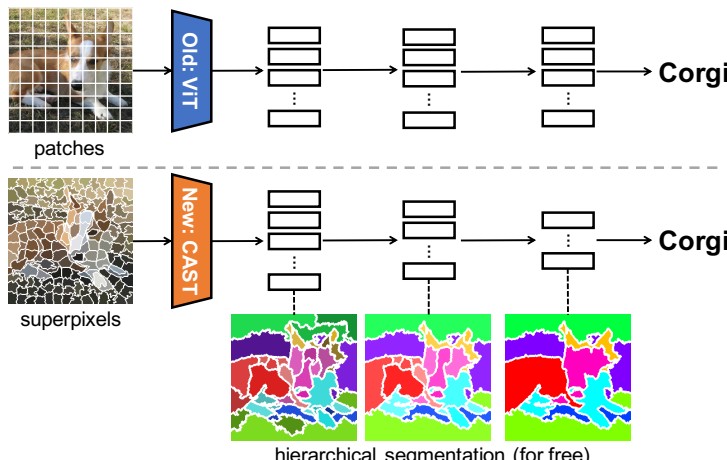

Figure 16: ViT takes patch tokens as inputs and maintains the same large number of tokens through all encoder blocks, whereas our CAST takes segment tokens as inputs and hierarchically coarsens them into fewer and larger region tokens. These segment tokens adapt to the image and vary in shape.

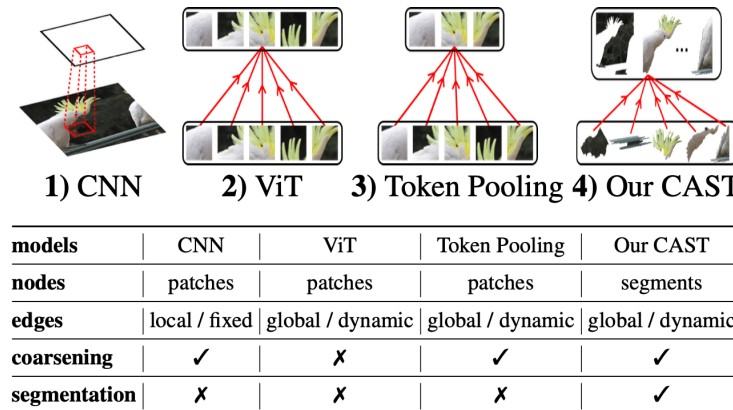

| models | CNN | ViT | Token Pooling | Our CAST |
|---|---|---|---|---|
| nodes | patches | patches | patches | segments |
| edges | local / fixed | global / dynamic | global / dynamic | global / dynamic |
| coarsening | ✓ | ✗ | ✓ | ✓ |
| segmentation | ✗ | ✗ | ✗ | ✓ |

Figure 17: We compare different models based on what they operate on and how they extract local-to-global information. CNN computes features on a grid and handles global information through spatial downsampling. ViT takes regularly shaped patches as inputs and updates features using attention. Token Pooling subsamples tokens based on their significance scores. Our CAST takes segment tokens as inputs and coarsens them into larger region tokens, which adapt to the image.

### E.3 OPEN-VOCABULARY SEGMENTATION

We use the open-vocabulary segmentation method OVSeg (Liang et al., 2023) to derive part-to-whole predictions from provided segments. OVSeg employs the CLIP (Radford et al., 2021) classifier on a masked image containing only the segment of interest, with other areas masked in gray. An additional "Background" class is introduced alongside the given foreground labels. For instance, PartImageNet (He et al., 2022) consists of 11 objects (e.g., Biped) and 40 parts (e.g., Biped Head), necessitating 12-way and 41-way classification, respectively. We predict category and object from the coarse segments and parts from fine segments. We use the ViT-B-16 model with pretrained weights released by OpenAI and do not fine-tune it with masked images.

CAST and ViT segments inherently have a hierarchy, so we restrict the part recognition vocabulary based on the object class of the parent segment. In contrast, SAM segments lack this hierarchy, necessitating a workaround to constrain their parent class. The specific procedures for each case are explained below. We applied GaussianBlur with kernel size 7×7 to smooth the segmentations.

**CAST (and ViT).** CAST progressively clusters segments as the layers advance, resulting in multiple levels of segments. In this context, we utilize the highest level (level 4) with 8 segments for object recognition and its child (level 3) with 16 segments for part recognition. We employ the same strategy for ViT, where the segments are generated using K-means clustering over embeddings at the same level. Thus, both CAST and ViT segments enforce a consistent hierarchy.

**SAM.** Segment Anything (SAM) (Kirillov et al., 2023) provides segments at different levels but does not consider their hierarchy. Specifically, SAM generates 64 segments with varying levels to parse the original image when no conditions are given. Here, we begin by sorting the segments in descending order of area, implying that the former ones tend to represent objects, while the latter ones tend to represent parts. We start with an empty semantic mask and iteractively fill the mask by processing the sorted segments to obtain the semantic segmentation.

While following the same procedure, an important distinction arises between object and part recognition. In object recognition, we skip updating the pixel if it is already filled since the earlier object segment has assigned it. Conversely, for part recognition, we override the pixel with the new part segment, enabling OVSeg to consider the fine-grained segments generated by SAM. We tested various approaches, and this strategy yielded the best results.

### E.4 COMPARISON WITH HIERARCHICAL SEGMENT GROUPING

Hierarchical Segment Grouping (HSG) (Ke et al., 2022) is an unsupervised hierarchical semantic segmentation framework that induces grouping hierarchy based on pixel feature similarity. Our CAST is similar to HSG in the sense of inferring hierarchical groupings in terms of merging segments progressively. Both methods predict soft assigned probabilities to map a fine-grained segment to a coarse-grained region at the next level. Multi-scale image segmentation is induced by chaining segment assignments across all levels.

However, our CAST differs from HSG in several aspects. **1)** HSG is designed specifically for unsupervised semantic segmentation. In stark contrast, our CAST is a general backbone architecture for different downstream tasks, e.g. classification, image segmentation, and detection. In addition, our model can be trained both supervisedly and unsupervisedly. **2)** Our hierarchical image segmentation is directly involved in multi-scale feature extraction. From the early stage of our model, every token features correspond to an image segment at different scales. In contrast, HSG builds atop CNN backbones, segmentations are only inferred at the final stage of the model. **3)** Our CAST demonstrates superior model efficiency by involving hierarchical segmentation and image recognition jointly. Segmentation is induced directly from the model architecture. HSG requires additional transformer modules to infer image segmentation.

### E.5 GENERATING SEGMENTS FROM VIT

We generate segments from ViT by applying K-Means clustering to the final output tokens. We bilinearly upscale the feature map and then apply K-Means to the pixel-level features to align the segments with the input image resolution. Similar to CAST, we cross-reference clustering assignments across different levels to achieve hierarchical image segmentation. To maintain a consistent cluster hierarchy, we iteratively run K-Means, reducing the number of clusters by grouping the clusters from the previous iteration into coarser clusters.

### E.6 TOKEN POOLING METHODS

We re-implemented previous token pooling methods: FINCH (Sarfraz et al., 2019), Token Pooling (Marin et al., 2021), CTM (Zeng et al., 2022), and ToMe (Bolya et al., 2023), for comparison with our proposed Graph Pooling. We use superpixel tokens for all methods, only modifying the token pooling layers. For FINCH and Token Pooling, we implemented their clustering algorithms. For TCFormer, we adapted Clustering-based Token Merge (CTM) module from the official codebase[1], removing the convolutional layer to apply it to our superpixel tokens. For ToMe, we used the official codebase[2] and reduced the tokens per layer to 16 to match the latency of other methods.

---

[1] https://github.com/zengwang430521/TCFormer
[2] https://github.com/facebookresearch/ToMe

# F ADDITIONAL EXPERIMENTAL DETAILS

## F.1 SUMMARY OF MODELS USED IN EXPERIMENTS

We pre-train our CAST model self-supervisedly and supervisedly on ImageNet and COCO dataset. Our experiments use different model architectures and pre-training objectives. For clarification, we summarize the backbone models used in each experiment in Table 10.

Table 10: Pre-trained backbone models used in each experiment.

| Experiment | Pre-training objective | Model |
|---|---|---|
| Hierarchical segmentation: ImageNet (Fig. 5) | self-supervised: IN-1K | CAST-S |
| Hierarchical segmentation: PartImageNet (Fig. 6 & Table 1) | self-supervised: IN-1K | CAST-B |
| Hierarchical segmentation: DensePose (Fig. 7 & Table 2) | self-supervised: COCO | CAST-S |
| Semantic segmentation: VOC (Table 3 a) | self-supervised: COCO | CAST-S |
| Semantic segmentation: ADE20K (Table 3 b) | supervised: IN-1K | CAST-S |
| Semantic segmentation: Pascal Context (Table 3 b) | supervised: IN-1K | CAST-S |
| Classification: ImageNet (Table 4) | self-supervised: IN-1K | CAST-S |
| Test-time adaptation: PartImageNet (Fig. 9 & Fig. 15) | self-supervised: IN-1K | CAST-B |
| Object-centric attention: VOC (Fig. 12) | self-supervised: IN-1K | CAST-S |

## F.2 CLASSIFICATION AND SEGMENTATION DATASETS

**Classification Datasets**. **ImageNet** (Deng et al., 2009) is a generic image classification dataset, annotated with $1,000$ object categories (IN-1K). The training and validation set includes 1.28M and 50K images, respectively. We follow Tian et al. (2020) to subsample 100 object categories to create IN-100. The subset consists of 127K and 5K images for training and testing.

**Segmentation Datasets**. **1) Pascal VOC 2012** (Everingham et al., 2010) is an object-centric semantic segmentation dataset that contains 20 object categories and a background class. We use the augmented training set (Hariharan et al., 2011) with $10,582$ images and the validation set with $1,449$ images. **2) MSCOCO** (Lin et al., 2014) is a generic scene dataset with complex contexts and include more objects in each image (7.3 vs. 2.3) than VOC. Following Van Gansbeke et al. (2021), we train on $118,287$ images of *train2017* split and test on the VOC validation set. **3) ADE20K** Zhou et al. (2019) is a complex scene dataset, annotated with 150 object categories. The dataset includes $20,210$ and $2,000$ images for training and validation. **4) Pascal Context** is also a scene dataset with $4,996$ and $5,104$ images for training and validation, labelled with 59 object categories and a background class.

## F.3 IMAGE RESOLUTION AND NUMBER OF TOKENS

We closely follow (Chen et al., 2021; Touvron et al., 2021; Van Gansbeke et al., 2021; Caron et al., 2021) to set up image resolution for classification and segmentation. For ViT baselines, on ImageNet, we set crop_size to 224, resulting in 196 input patch tokens. On VOC, we use $512 \times 512$ input images with corresponding 1024 patch tokens for semantic segmentation. We follow the same setup as (Strudel et al., 2021) for experiments on ADE20K and Pascal Context. On ADE20K, we use a $512 \times 512$ sliding window and set the stride to 512. On Pascal Context, we use $480 \times 480$ sliding window and set the stride to 320. For our CAST, we adopt the same image resolution settings and adjust the granularity of superpixels to match the number of input tokens as ViT baselines.

### F.4 Hyper-parameters for Training

We list the hyper-parameters for training using MoCo and DeiT framework in Table 11 and Table 12. We mostly follow the default hyper-parameters used in each framework. We set the same batch_size and total_epochs as our CAST and ViT baselines. All the experiments are conducted with either eight TitanX (12 GB) or two A100 (45 GB) Nvidia graphic cards.

Table 11: Hyper-parameters for training our CAST, ViT, and Swin on IN-100, IN-1K, and COCO dataset. Due to computating limitations, we adapt batch_size and total_epoch in our experiments. Otherwise, we follow mostly the same set up as MoCo (Chen et al., 2021).

| Parameter | IN-100 | IN-1K | COCO |
|---|---|---|---|
| batch_size | 256 | 256 | 256 |
| crop_size | 224 | 224 | 224 |
| learning_rate | $1.5e^{-4}$ | $1.5e^{-4}$ | $1.5e^{-4}$ |
| weight_decay | 0.1 | 0.1 | 0.1 |
| momentum | 0.9 | 0.9 | 0.9 |
| total_epochs | 200 | 100 | 400 |
| warmup_epochs | 20 | 10 | 40 |
| optimizer | Adam | Adam | Adam |
| learning_rate_policy | Cosine decay | Cosine decay | Cosine decay |
| MOCO : temperature | 0.2 | 0.2 | 0.2 |
| MOCO : output_dimension | 256 | 256 | 256 |
| MOCO : momentum_coefficients | 0.99 | 0.99 | 0.99 |
| MOCO : MLP hidden dimension | 4096 | 4096 | 4096 |
| ViT: # Tokens | $[196]_{\times 11}$ | | |
| CAST-S/B: # Tokens | $[196]_{\times 3}, [64]_{\times 3}, [32]_{\times 3}, [16]_{\times 2}$ | | |

Table 12: Hyper-parameters for training our CAST and ViT on IN-1K dataset. We follow mostly the same set up as DeiT (Touvron et al., 2021).

| Parameter | IN-1K |
|---|---|
| batch_size | 1024 |
| crop_size | 224 |
| learning_rate | $5e^{-4}$ |
| weight_decay | 0.05 |
| momentum | 0.9 |
| total_epochs | 300 |
| warmup_epochs | 5 |
| warmup_learning_rate | $1e^{-6}$ |
| optimizer | Adam |
| learning_rate_policy | Cosine decay |
| augmentation | RandAug(9, 0.5) (Cubuk et al., 2020) |
| label_smoothing (Szegedy et al., 2016) | 0.1 |
| mixup (Zhang et al., 2017) | 0.8 |
| cutmix (Yun et al., 2019) | 1.0 |
| ViT: # Tokens | $[196]_{\times 11}$ |
| CAST-S: # Tokens | $[196]_{\times 3}, [64]_{\times 3}, [32]_{\times 3}, [16]_{\times 2}$ |

### F.5 INFERENCE AND EVALUATION ON IMAGENET AND VOC

For image classification on ImageNet (Fig. 18), we apply the linear probing procedure for evaluation. For semantic segmentation on VOC (Fig. 19), we use the segment retrieval and transfer learning procedure to evaluate model performance.

**Image classification: linear probing.** For unsupervised classification, we follow MoCo-v3 (Chen et al., 2021) to evaluate image-wise discrimination model performance using a linear probing protocol. We freeze the trained model weights and replace the 3-layer MLP head with a randomly initialized linear projection layer as classifier. We train the linear classifier with ground-truth labels and report the top-1 accuracy. Following Chen et al. (2021), we train the linear classifier with 90 epochs on ImageNet dataset. We set $\mathrm{momentum}$ to 0.9 and $\mathrm{weight\_decay}$ to 0 for all experiments. On IN-1K, we set $\mathrm{batch\_size}$ to 1024, $\mathrm{learning\_rate}$ to 30; on IN-100, we set $\mathrm{batch\_size}$ to 256, $\mathrm{learning\_rate}$ to 0.8. SGD is used as the optimizer.

**Semantic segmentation: segment retrieval.** We follow Hwang et al. (2019); Van Gansbeke et al. (2021); Ke et al. (2022) to evaluate semantic segmentation using segment retrieval. We partition an image into several segments and conduct nearest neighbor search to predict the label for each segment. We assign the majority labels from the 20 retrieved segments.

For ViT baselines, we apply the MLP head on each token to generate unit-length output features and upsample the feature maps to the original resolution of the input image. Followed by spherical K-Means clustering algorithm, we partition the image into 40 segments using the output features.

Our CAST does not require additional upsampling and K-Means clustering. For segmentation, our model follows Hypercolumn design (Hariharan et al., 2015) to unpool and fuse multi-level segment tokens. Our model generates the same number of output tokens as the superpixels. We gather pixel features from output tokens based on the superpixel index. Without the need for spherical K-Means clustering, our CAST predicts an image segmentation using the graph pooling modules. We compute normalized segment features according to such image segmentation.

**Semantic segmentation: transfer learning.** We follow Van Gansbeke et al. (2021) to evaluate model performance using transfer learning protocol. All models are unsupervisedly trained on MSCOCO, and supervisedly fine-tuned on Pascal VOC. We replace the 3-layer MLP head with 2-layer $1 \times 1$ convolutional layers. For training ViT baselines, we upscale patch tokens back to the input image resolution using bilinear interpolation. For training our CAST, we scatter segment tokens to per-pixel features using superpixel indices. We use per-pixel ground-truth labels for training both methods. We set the training steps to 30K, $\mathrm{learning\_rate}$ to 0.003, $\mathrm{weight\_decay}$ to 0.0001, $\mathrm{batch\_size}$ to 16, $\mathrm{crop\_size}$ to 512. Following Chen et al. (2016), we adopt poly learning rate policy by multiplying base learning rate by $1 - \frac{iter}{max\_iter}^{0.9}$. We adopt the SGD optimizer. We use only a single-scale image for inference.

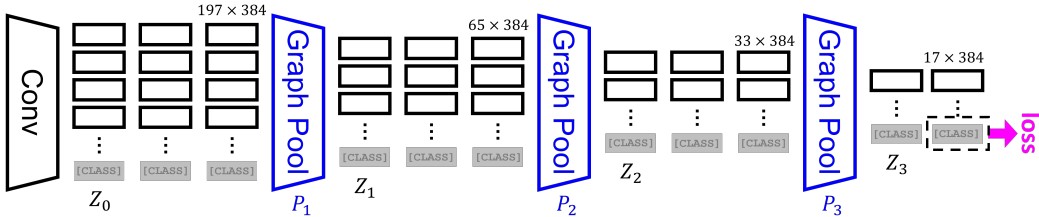

Figure 18: Detailed model architecture for classification. Our CAST aggregates segment tokens by average pooling convolutional features within each superpixel, contextualize segment tokens with transoformer encoder blocks, and coarsen them into fewer region tokens with Graph Pooling module.

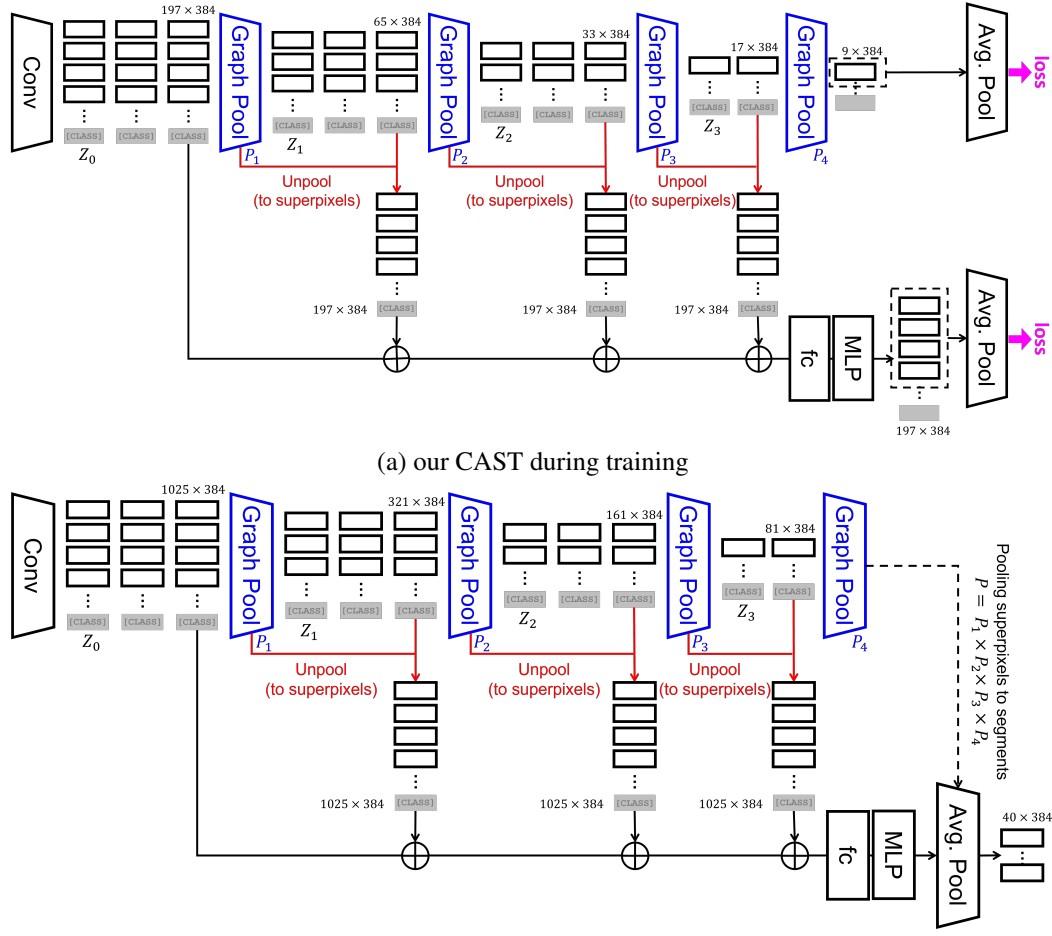

(a) our CAST during training

(b) our CAST during inference for nearest segment retrievals

Figure 19: Detailed model architecture for semantic segmentation. Our CAST aggregates segment tokens by average pooling convolutional features within each superpixel, contextualize segment tokens with transoformer encoder blocks, and coarsen them into fewer region tokens with Graph Pooling module. **(a)** Our CAST-S during training. **(b)** Our CAST-S during inference for segment retrievals. We unpool coarsened segment tokens based on the grouping index w.r.t input superpixels. We concatenate ($\oplus$) unpooled segment tokens across multiple scales and fuse them using a fully_connected layer, followed by the 3-layer MLP head. For transfer learning, we replace the 3-layer MLP head with 2-layer $1 \times 1$ convolutional layers atop the fused tokens. We also remove the final average pooling layer. For segment retrievals, we learn an additional Graph Pooling module to predict coarse segmentations and average pool tokens as outputs for nearest-neighbor retrievals.

# G ADDITIONAL VISUALIZATIONS

## G.1 VISUAL RESULTS ON HIERARCHICAL SEGMENTATION

We present additional visualizations of hierarchical segmentations induced by ViT and our CAST (Fig. 20). CAST captures image boundaries and semantics more precisely.

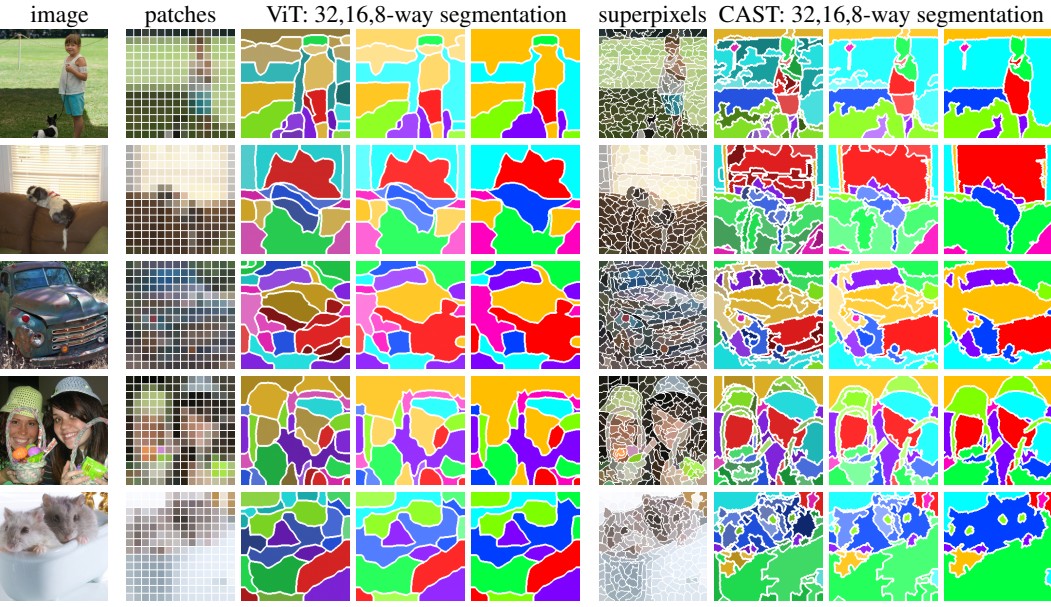

Figure 20: Additional visual results on hierarchical segmentation show that our model captures better contours and the wholes segmentation than ViT.

## G.2 VISUAL RESULTS ON SEMANTIC SEGMENTATION

We present more visualization results of before and after fine-tuned semantic segmentation on VOC (Fig. 21). Compared to ViT baselines, our model produces more accurate segmentations. Remarkably, our results align with object contours precisely without the need of additional CRF post-processing.

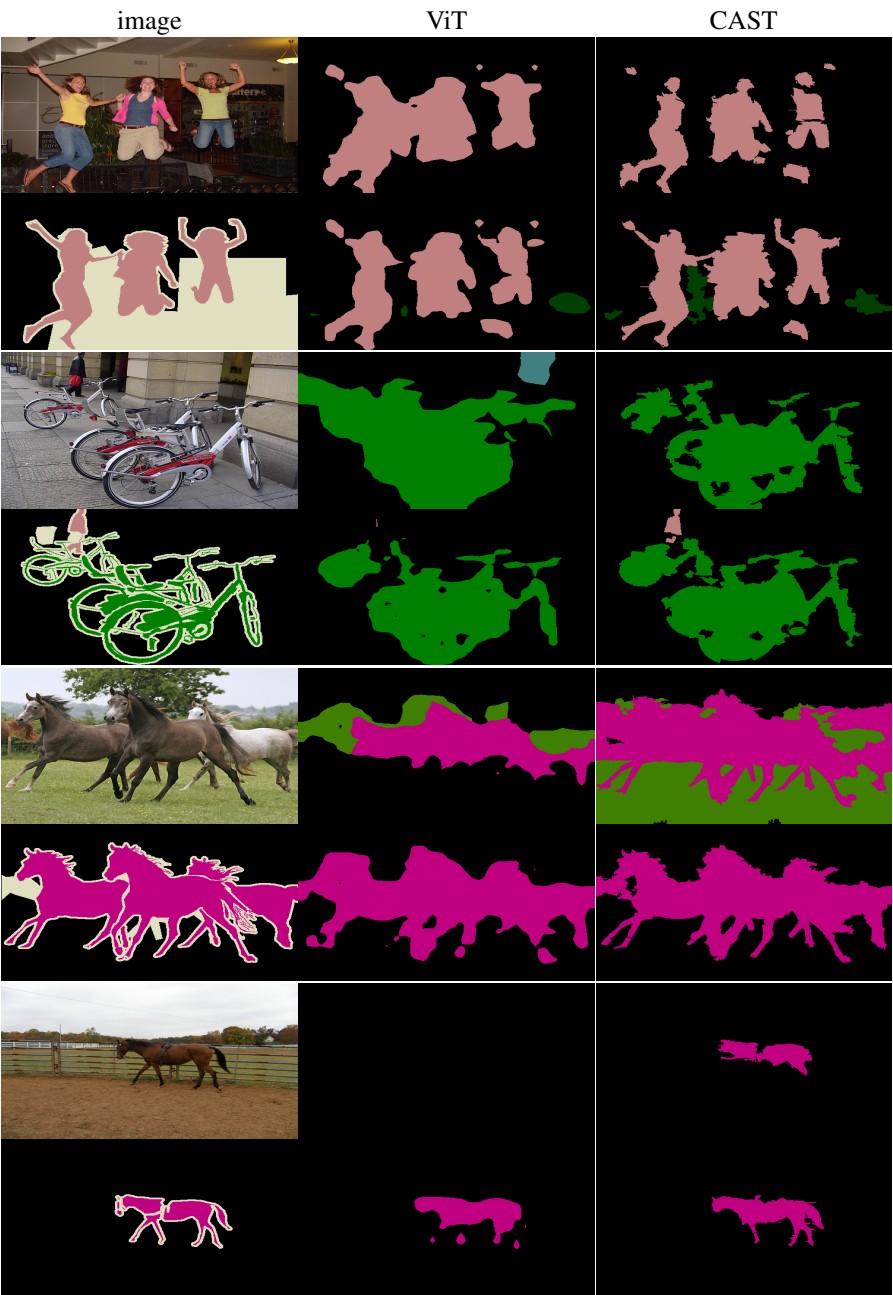

Figure 21: Our model induce much more precise segmentation than patch tokens. Segmentations are predicted based on segment-wise nearest neighbor retrievals (row 1 images) and fine-tuned models (row 2 images). Using segment, not patch, tokens improves our predicted segmentations by a large margin. Notably, our method explicitly produces segmentations without the need of additional K-Means clustering for segment retrievals.

### G.3 VISUAL RESULTS ON FIGURE-GROUND SEGMENTATION

We present more visual results of figure-ground segmentations generated from multi-head attention maps on VOC. Our CAST attends to foreground semantics more precisely than ViT, and the segmentations preserve object boundaries more accurately. See Fig. 22.

CAST (top) vs. ViT (bottom)

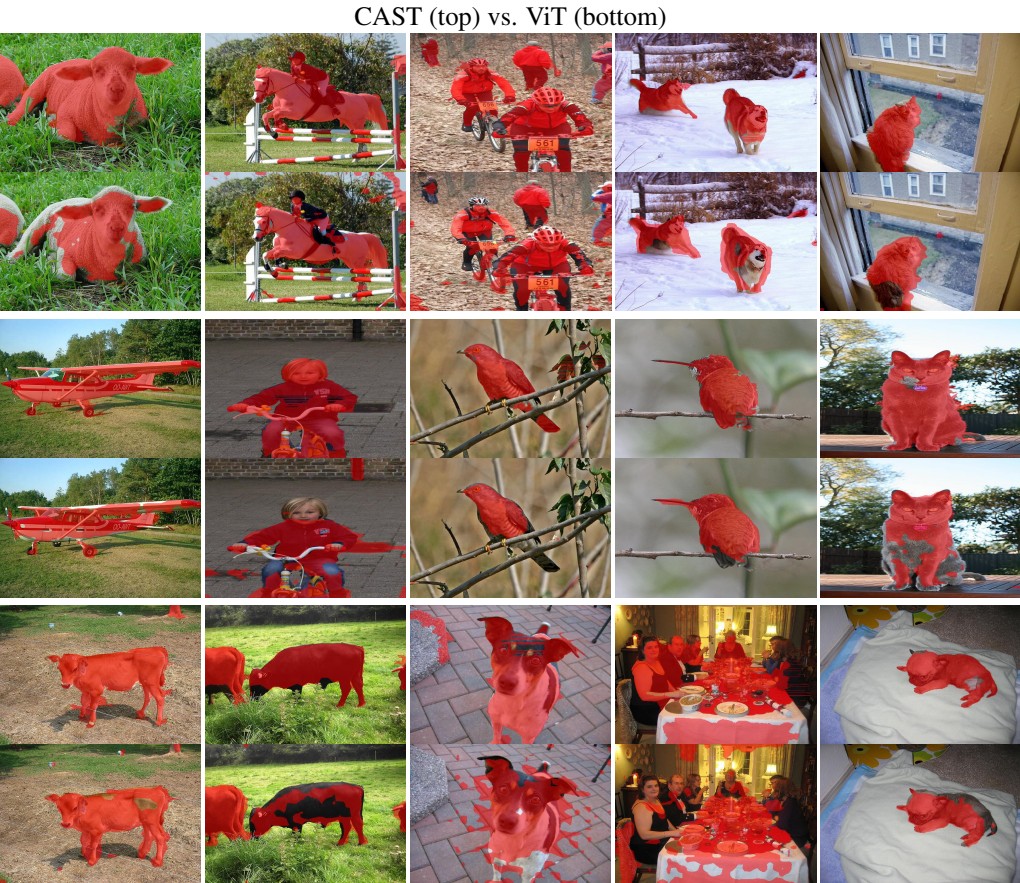

Figure 22: Our CAST (top row) attends to foreground semantics more precisely than ViT (bottom row) and DINO (Caron et al., 2021) on VOC. We adopt the same procedure as DINO to generate foreground segmentation masks from latent multi-head attention maps. All models are trained on IN-1K dataset from scratch. Our CAST and ViT are trained based on MoCo-v3 (Chen et al., 2021).

## G.4   VISUAL RESULTS ON MULTI-HEAD ATTENTION MAPS

We visualize the multi-head attention maps of the [CLASS] token to all the other segment tokens in our vision transformer. As the [CLASS] token is optimized for image-wise discrimination, such attention maps indicate the most informative groupings of segments that will induce the most discriminative image-wise representations. We visualize the same attention maps used to generate the figure-ground segmentation, which are the ones in the $9^{th}$ transformer encoder block. The layer takes 32 coarsened segment tokens as inputs, resulting in 12 heads of $32 \times 32$ attention maps. We follow the same procedure as DINO (Caron et al., 2021) to display the binarized attention maps. The threshold is adjusted to keep 60% of the mass. See Caron et al. (2021) for more details.

As shown in Fig. 23, our attention maps reveal parts-of-the-whole information of the image. We observe that the same object parts are together attended in the same attention head, e.g. face vs. ears vs. nose of the dog. It indicates that image-wise recognition requires parts-of-the-whole information. Additionally, our model carries segment, not patch, tokens through the layers, resulting in attention maps better aligned with object boundaries.

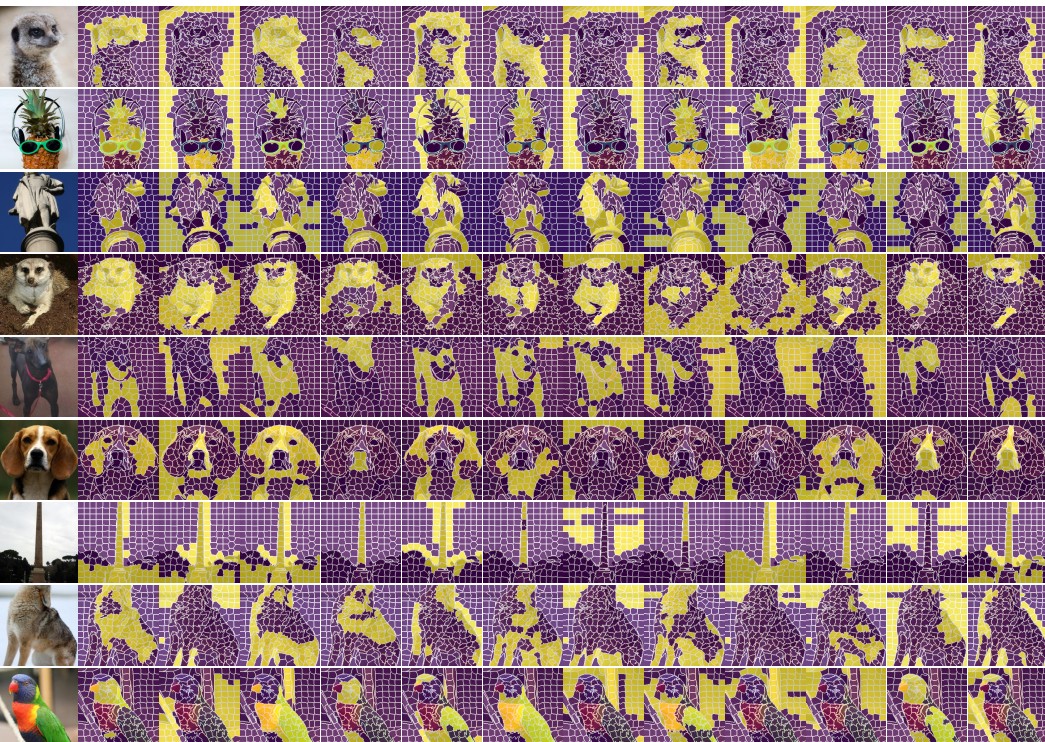

Figure 23: Our multi-head attention maps reveal parts-of-the-whole information of the image on IN-100. **From left to right:** input images and corresponding 12 heads of attention maps of the [CLASS] token to all the other segments. We follow DINO (Caron et al., 2021) to binarize attention maps. We show that the same object parts are together attended in the same head, e.g. face vs. ears vs. nose of the dog. Our model takes segment tokens, resulting in attention maps better aligned with object boundaries.

