# OpenReview forum: "Learning Hierarchical Image Segmentation For Recognition and By Recognition"
_ICLR.cc/2024/Conference — ICLR 2024 spotlight_

### Official Review · Reviewer_GfNL · 2023-10-29

**Soundness:** 4 excellent
**Presentation:** 3 good
**Contribution:** 4 excellent
**Rating:** 8
**Confidence:** 4

**Summary:**

This paper introduces CAST (Concurrently Adaptive Segmentation Tokens).
This builds on existing work where superpixels are used as tokens in a
vision transformer. The new contribution is that graph pooling is used to
adaptively cluster segments to create a hierarchy, allowing segmentation (and
classification) at a coarse and fine-grained level.

SEEDS is used to define superpixels via over-segmentation. A CNN produces
initial features on the original pixel regular grid. The mean of CNN features
under a superpixel define the feature-token for that superpixel. Positional
encodings are created in a similar way.

Superpixels are used as leafs of a tree, which is adaptively defined by graph
pooling. This tree has L layers, and starting from the leafs, the next layer is
defined by first assuming that there are C coarser segments at the next level,
and then for each token the network predicts a probability for which coarser
segment each finer segment should be assigned to. The argmax of the probability
is the assignment. The number of coarser segments is defined adaptively via the
Farthest Point Sampling algorithm.


Results are reported on:

* PartImageNet, where CAST is compared to a baseline ViT and SAM-B, where is shows a significant performance improvement over the baseline in most cases and also with lower computational costs.

* DensePose Human part segmentation, and is comparable to the HSG baseline.

* Pascal VOC Segmentation - where CAST and ViT are trained on COCO. Ablation studies are
  done here. CAST outperforms the baseline in all reported cases.

* Pascal VOC classification - where CAST is compared to ViT and Swin using a linear probing evaluation.

**Strengths:**

The algorithm design is well motivated.

The technique is tested against strong baselines, and shows benefits in both quality and computational metrics.

Design choices are ablated.

Additional details and failure cases are given in the appendix.

**Weaknesses:**

The tone is a bit to strong and unscientific at times. Value judgements are
placed on things, instead of performing comparisons and reporting results. I
recommend rewording in certain areas:

> We assert that our design is the proper approach for true vision transformer, distinct from the text-inspired ViT.

There is no "proper" approach. You can assert it might be a more natural choice in some circumstances, but in the instance where superpixel segmentation fails, then this is very much not the correct approach.

> CAST not only discovers the hierarchy but also enhances flat semantic segmentation, indicating its superiority in learning dense representations compared to ViT.

Superiority is too strong. It might be ok, but given the overstated tone in the rest of the paper, I think: "indicating that it learns richer dense representations" might be a better wording.

> Sec. 4.4. We conducted an ablation study on our design choices, confirming that CAST is better
than previous token pooling approaches for ViT (Marin et al., 2021; Sarfraz et al., 2019).

It's not confirmed. You made a measurement that contributed evidence towards
the idea. Confirming is something that someone who replicates your work will
do. I recommend changing the word "confirmed" to measured in most places.

**Questions:**

> The resulting input segment features are defined as Z0 = [Xclass; Xs] + Epos

I've seen people simply adding positional encodings to the token features directly rather than concatenating them as additional feature channels. I must have missed the work that introduced / ablated this decision. Is there a reference you can point me to on this?


> We initiate the next-level coarse regions by sampling centroids from segment tokens Zl−1 at level l − 1 and compute Pl based on token similarity, with C representing the sampled centroid indices.

Are these centroids in position space or feature space?


> All models are self-supervisedly trained on IN-100 and evaluated using linear probing.

Is there a reference for this? I'm not familiar with this and it seems like a traditional imagenet-style classification benchmark would be a better comparison where there was a defined groundtruth.

---

> ### Author Response · Authors · 2023-11-19
> **Response to Reviewer GfNL**
>
> Dear reviewer GfNL,
>
> Thank you for your valuable feedback and comments. We appreciate your remarks on the motivation, method design, and experimental results of our work. We will address your concerns and questions in the response below.
>
> ---
> **[W1] Tone is a bit too strong**
>
> Thank you for your suggestions. We revised our manuscript accordingly.
>
> ---
> **[Q1] How to incorporate positional encodings?**
>
> We added the positional encodings instead of concatenating them like standard ViTs. Please note that $E_\text{pos}$ is added ($+$) to the tokens in the equation on page 4 of our paper.
>
> ---
> **[Q2] Centroids from position or feature space?**
>
> The centroids are sampled based on token similarity in the feature space, though it also reflects the position information through positional encodings. We have clarified this on page 5 of our revised manuscript.
>
> ---
> **[Q3] Reference for linear probing on IN-100**
>
> Linear probing is a widely used protocol for evaluating self-supervised learning methods [1-3]. It involves training a linear classifier on top of learned representations to measure classification accuracy. Prior works [4-5] used ImageNet-100 for linear probing, where the top-1 accuracy (~80%) in Table 1 matches that in Table 4 of our paper. The detailed procedure has been explained in Appx. C.4 of our manuscript.
>
> [1] He et al. Momentum Contrast for Unsupervised Visual Representation Learning. CVPR 2020.\
> [2] Chen et al. A Simple Framework for Contrastive Learning of Visual Representations. ICML 2020.\
> [3] Caron et al. Emerging Properties in Self-Supervised Vision Transformers. ICCV 2021.\
> [4] Xiao et al. What Should Not Be Contrastive in Contrastive Learning. ICLR 2021.\
> [5] Costa at al, Solo-learn: A Library of Self-supervised Methods for Visual Representation Learning. JMLR 2022.
>
> ---
> Please let us know if you have any further concerns.
>
> Sincerely,\
> Authors

---

> > ### Comment · Reviewer_GfNL · 2023-11-22
> >
> > Thank you, I've updated my rating of presentation from 2 to 3. I like the paper and will stick with the score of 8.

---

### Official Review · Reviewer_hsyp · 2023-10-30

**Soundness:** 3 good
**Presentation:** 4 excellent
**Contribution:** 3 good
**Rating:** 6
**Confidence:** 4

**Summary:**

This work addresses image classification and hierarchical image segmentation. No pixel-wise ground truth is provided in training for segmentation, so the proposed ViT-based approach, CAST, is trained by self-supervision and supervised image classification. CAST replaces the fix-grid patches with superpixels, and uses a graph pooling to aggregate features to sampled cluster centroids. The resulting graph-pooled superpixel tokens are aimed at better incorporating the local-global hierarchical semantics, and can produce image segmentation. CAST is evaluated on multiple datasets, and outperforms SOTA.

**Strengths:**

This paper is well-written and easy to follow. The experiments are comprehensive, and the reported results are promising.

The motivation for using the superpixel token to replace the fixed-grid patch token in ViT is clear and interesting. As the initial superpixels are estimated based on low-level visual cues, the graph-pooled superpixel tokens incorporate appearance and shape details, as well as local-global hierarchical relationships. This seems to help give sharper segmentation masks than in related work.

**Weaknesses:**

Overall novelty seems limited. The proposed graph-pooling for aggregating fine-level tokens into coarse-level tokens is similar to the Multi-stage Token Aggregation in TCFormer (Zeng et al 2022) and the Token Merging in ToMe (Bolya et al 2023). A comparison of CAST with these two methods is missing.

The method increases complexity relative to using the fixed-grid of patches, since it requires estimation of superpixels and sampling the clusters. Performance seems to be critically dependent on the quality of superpixels and the cluster sampling methods. The paper poorly tests performance wrt varying quality of superpixels and cluster sampling. It is unsatisfying that these two components are not learnable, and that they are not end-to-end integrated with the rest of CAST.

**Questions:**

What is the "MLP ratio" in section 4.4 and Table 6.c? Above equation 3, it shows f(a,b) = a + MLP(b), but there is no coefficient there.

---

> ### Author Response · Authors · 2023-11-19
> **Response to Reviewer hsyp**
>
> Dear reviewer hsyp,
>
> Thank you for your valuable feedback and comments. We appreciate your remarks on the motivation, experimental results, and presentation of our work. We will address your concerns and questions in the response below.
>
> ---
> **[W1] Technical novelty over token pooling**
>
> Our graph pooling is just a device for implementing our concept of hierarchical segmentation, and any token pooling method can be used. We emphasize that our goal is to develop a framework that integrates segmentation into the recognition process, unlike previous token pooling methods that focus on ViT efficiency. As a reminder, we have discussed this in the related section, properly mentioning both suggested works.
>
> We employed graph pooling because it performed the best in our experiments. In particular, we compared various token pooling algorithms in Table 6a, including Token Pooling (Marin et al., 2021) and FINCH (Sarfraz et al., 2019). Additionally, following your suggestion, we conducted further comparisons with state-of-the-art methods, TCFormer (Zeng et al., 2022) and ToMe (Bolya et al., 2023), using the same setup as in Table 6a. For TCFormer, we modified the Clustering-based Token Merge (CTM) module by removing the convolutional layer to apply it to our superpixel tokens. For ToMe, we reduced the number of tokens per layer to 16 to align with the latency of other methods. We stated the implementation details in Appx. A.5.
>
> The table below presents the results, including both the values from Table 6a and the new results. While both CTM and ToMe perform well, our graph pooling outperforms them. We have updated the table in our revised manuscript.
>
> | Pooling Method       | Accuracy |
> |----------------------|----------|
> | Graph Pooling        | **79.9** |
> | Random Sampling      | 55.8     |
> | K-Means              | 73.9     |
> | K-Medoids            | 72.3     |
> | FINCH                | 63.3     |
> | Token Pooling        | 75.8     |
> | CTM (new results)   | 72.2     |
> | ToMe (new results)   | 78.1     |
>
> ---
> **[W2] Ablation study on superpixel and cluster sampling methods**
>
> *Superpixels.*
> We used SEEDS superpixels for simplicity, as they performed well. However, we note that CAST also performs well with other reasonable superpixel methods. To verify this, we conducted an additional ablation study on superpixel methods. Specifically, we train CAST-S models using SEEDS (Bergh et al., 2012) and SLIC (Achanta et al., 2012) superpixels with MoCo objectives. We report the linear probing accuracy on ImageNet-1K using the models trained on ImageNet-1K and mIoU on Pascal VOC using the models trained on COCO, both before (left) and after (right) fine-tuning.
>
> The table below compares the classification and segmentation performance of SEEDS and SLIC. Both superpixels performed reasonably well, with SEEDS outperforming SLIC by capturing better boundaries. We included deeper discussions and visual examples of superpixels in Appx. B.8 of our revised manuscript.
>
> | Superpixel | Classification | Segmentation |
> |--------|--------|-------|
> | SEEDS  | 67.0   | 38.4/67.6 |
> | SLIC   | 65.6   | 37.7/65.7 |
>
> Nevertheless, we acknowledge that CAST indeed depends on the quality of superpixels. We discussed this limitation in Appx. B.7, illustrating that superpixels may not capture thin structures like light poles. Addressing this issue, perhaps by jointly learning superpixels, would be an interesting future direction.
>
> *Cluster sampling.*
> Our cluster sampling is jointly learned with feature development in CAST. On one hand, our cluster sampling and assignments are determined by the learned feature distances. On the other hand, our clusterings facilitate the final recognition process, guiding feature learning. Thus, cluster sampling and feature learning mutually influence each other in CAST. We note that Farthest Point Sampling (FPS) is just one natural method for partitioning features into clusters, and other reasonable clustering methods that encourage maximum margins between clusters would perform similarly well.
>
> ---
> **[Q1] Definition of the MLP ratio?**
>
> The MLP ratio refers to the multiplier for the dimension of the MLP used in the self-attention block within our graph pooling module. We clarified this in our revised manuscript.
>
> ---
> Please let us know if you have any further concerns.
>
> Sincerely,\
> Authors

---

### Official Review · Reviewer_RnRE · 2023-11-02

**Soundness:** 4 excellent
**Presentation:** 4 excellent
**Contribution:** 4 excellent
**Rating:** 10
**Confidence:** 4

**Summary:**

This paper presents a novel method that adopts superpixel as the token for vision transformer. The superpixel tokens are hierarchically grouped together in each layer via graph pooling. This gives rise to a hierarchical segmentation result  using only image classification label. Experimental results demonstrate the usefulness of the proposed method.

**Strengths:**

- Using superpixel tokens is more appropriate for vision transformers. The proposed method delivers a hierarchical segmentation result using only image-level annotation.  I really appreciate the concept of using hierarchical segmentations as the tokens in different layers. In particular, the proposed method starts with a finest segmentation given by superpixels, and learns automatically the hierarchical segmentation. In my opinion, using different regions in hierarchical segmentation is a more natural and powerful way of image tokenization.
- The experimental results are quite convincing. The generated hierarchical segmentation is visually impressive. As depicted in Table 1 and 3, the proposed method significantly outperforms some baseline methods (e.g., the powerful segment anything model), while being also efficient.

**Weaknesses:**

- The runtime analysis is missing. Compared with the vanilla ViT, the proposed method involves superpixel generation and graph pooling. Does this require much more extra runtime?
- Ablation study of using different superpixel methods. Is this proposed method sensitive to the finest segmentation ?

**Questions:**

Does this require much more extra runtime compared with classical ViT.
Is this proposed method sensitive to the finest segmentation ?

---

> ### Author Response · Authors · 2023-11-19
> **Response to Reviewer RnRE**
>
> Dear reviewer RnRE,
>
> Thank you for your valuable feedback and comments. We appreciate your remarks on the motivation, method design, and experimental results of our work. We will address your concerns and questions in the response below.
>
> ---
> **[W1] Runtime analysis**
>
> The computational cost of superpixel generation and graph pooling can be reduced by the decreased number of tokens required for computing self-attention. This advantage becomes more pronounced when using larger models, where self-attention blocks dominate the entire cost. To validate this, we analyze the latency of model inference and superpixel generation. Our system comprises a 32GB Nvidia Titan V GPU card and two Intel(R) Xeon(R) CPU E5-2630 v4 processors, totaling 20 CPU cores. We utilize the PyTorch machine learning framework with 24 workers, a batch size of 64, and an image resolution set to 224x224.
>
> In our system, CAST-B achieves a lower average inference latency of 217 ms compared to ViT-B with 273 ms. SEEDS takes 73 ms to generate superpixels from the batch of images. However, we remark that the current SEEDS implementation is not fully optimized. Employing GPU implementation or parallelizing the process with more CPU cores can alleviate the bottleneck in superpixel generation. Furthermore, the cost of superpixel generation becomes less significant with larger models, which are commonly used in practice. We have included these discussions in Appx. B.1 of our revised manuscript.
>
> ---
> **[W2] Ablation study on superpixel methods**
>
> CAST performs well with reasonable superpixel methods. To verify this, we conducted an additional ablation study on superpixel methods. Specifically, we train CAST-S models using SEEDS (Bergh et al., 2012) and SLIC (Achanta et al., 2012) superpixels with MoCo objectives. We report the linear probing accuracy on ImageNet-1K using the models trained on ImageNet-1K and mIoU on Pascal VOC using the models trained on COCO, both before (left) and after (right) fine-tuning.
>
> The table below compares the classification and segmentation performance of SEEDS and SLIC. Both superpixels performed reasonably well, with SEEDS outperforming SLIC by capturing better boundaries. We included deeper discussions and visual examples of superpixels in Appx. B.8 of our revised manuscript.
>
> | Superpixel | Classification | Segmentation |
> |--------|--------|-------|
> | SEEDS  | 67.0   | 38.4/67.6 |
> | SLIC   | 65.6   | 37.7/65.7 |
>
> Nevertheless, we acknowledge that CAST indeed depends on the quality of superpixels. We discussed this limitation in Appx. B.7, illustrating that superpixels may not capture thin structures like light poles. Addressing this issue, perhaps by jointly learning superpixels, would be an interesting future direction.
>
> ---
> Please let us know if you have any further concerns.
>
> Sincerely,\
> Authors

---

### Official Review · Reviewer_5FXt · 2023-11-07

**Soundness:** 3 good
**Presentation:** 4 excellent
**Contribution:** 3 good
**Rating:** 8
**Confidence:** 3

**Summary:**

This paper presents CAST - a variant of ViTs that does hierarchical segmentation of inputs and tokens as part of its pipeline. The method is simple - input pixels are partitioned into super-pixels (in contrast to simple patches for ViTs) a convolutional network is applied over the input image and the resulting features are pooled across the superpixels (average pooling) to create the initial set of tokens.
From here the method works by passing the set of tokens through ViT blocks and between them graph pooling layers which group tokens into a coarser set of tokens by sampling centroids and measuring similarity between tokens and centroids.

The method is trained with a self supervised objective (MoCo, read out from the last layer) and demonstrated to work well in a variety of contexts.

**Strengths:**

I think this is an interesting paper with good motivation and a simple and effective implementation of core ideas.

* While simple, the use of super pixels and feature pooling to replace the awkward patch based embeddings of ViTs is a nice and original idea.
* The method is quite elegant and naturally lends itself to several use cases which are demonstrated in the paper in a convincing manner (mostly)
* Paper is well presented and well executed and was quite easy to follow.
* A highly applicable method which should be easy to use in several contexts so significant to the community.

**Weaknesses:**

I have some (relatively minor) concerns about parts of this work:

* The major thing which I find missing is more in-depth discussion of the role of super-pixels here - I feel the ablation table is missing one critical row which is a ViT with a super-pixel (+conv net) embedding layer instead of patch extraction. This is also a possible explanation to the results in Table 5. There is a chance that super-pixels are the source of most of the improvements in the paper (quantitatively, at least, clearly there would be no clear hierarchical structure in this case) and this is not discussed enough.

* The positional embedding pooling is a bit odd - super pixels can be with quite arbitrary shapes and there is a chance the resulting average PE over the super pixels would bare little connection to the actual "position" of the super pixel. Can the authors comment on that?

* While the authors claim there's a top-down element here (and arguably there is through training) I would argue it is not truly top-down at inference. The super pixels and segmentations do not depend and will not change based on what the top layers infer - the network has top-down pathway to inform them (other than gradients in training). This is of course fine, but it should be discussed.

I hope to see a discussion about the points above and I would be happy to raise my score if these are discussed and addressed to.

**Questions:**

More minor questions:

* How do segmentations are extracted from ViTs in the paper? it's not clear to me how the results in, say, Figure 4, were generated for ViT-S.

* Is the PE added to the CLS token as well?

---

> ### Author Response · Authors · 2023-11-19
> **Response to Reviewer 5FXt**
>
> Dear reviewer 5FXt,
>
> Thank you for your valuable feedback and comments. We appreciate your comments on the motivation, method design, presentation, and applicability of our work. We will address your concerns and questions in the response below.
>
> ---
> **[W1] Discussion on the role of superpixels**
>
> We have shown that superpixels are not the sole reason for improvements. Please refer to the discussion in Table 3a, which reports (non-hierarchical) semantic segmentation performance. Here, superpixels enhance segmentation by providing better boundaries, as you suggested. However, combining hierarchical pooling leads to even greater improvements, enabling the model to grasp the structure of entire objects. This supports that both superpixels and pooling are essential, even for applications that do not require explicit hierarchy. We have further clarified these points on page 8 of our revised manuscript.
>
> We have included the reorganized Table 3a below for your convenience. In this table, we report the mIoU (left) and boundary F-score (right) of segmentation before and after fine-tuning the models. Superpixels improve the original ViT by +5.1% in the boundary F-score. However, hierarchical pooling further enhances ViT + Superpixel by +6.8%.
>
> | Model | Before tuning | After tuning |
> |-------|-------|-------|
> | ViT | 30.9/16.1 | 65.8/40.7 |
> | ViT + Superpixel | 32.2/21.2 | 66.5/46.7 |
> | ViT + Superpixel + Hierarchy (CAST) | **38.4**/**27.0** | **67.6**/**48.1** |
>
> ---
> **[W2] Pooling over positional embeddings**
>
> Pooling over positional embeddings is as natural as pooling over token features, with the former characterizing spatial location and the latter characterizing appearance, which are two aspects of a segment. In fact, averaged positional embeddings serve as summary statistics for representing pooled segments, even when the corresponding location falls outside of a non-convex shaped segment pool. Prior works have also adopted a similar design choice. For example, SLIC (Achanta et al., 2012) calculates centroids by averaging LAB color features and XY locations within pixel clusters.
>
> ---
> **[W3] Top-down recognition during inference**
>
> Thank you for the insightful suggestion! As you noted, CAST only considers the top-down pathway during training, guided by the recognition objectives. This knowledge is encoded in the model and reflects the bottom-up pathway during inference. While this enables the model to learn the general top-down elements, the segmentations will not change based on what the top layers predict at inference time.
>
> To extend CAST by incorporating the top-down pathway during inference, we employ test-time adaptation (TTA) [1], specifically TENT [2], with the classifier trained on top of the CAST backbones. We apply TENT to each sample to adapt the model and maximize its prediction confidence. As a result, CAST refines its object segments to align with its initial belief: If this image depicts the predicted class, which parts contribute to this prediction?
>
> We have included the detailed approach and experimental results in Appx. D of our revised manuscript. We trained a dog vs. non-dog classifier on PartImageNet and evaluated the evolution of segmentation before and after TTA. Figure 16 showcases visual examples where TTA improves the segmentation of CAST by capturing object shapes more effectively (e.g., not missing legs in rows 1-3) while reducing attention to unnecessary details (e.g., window frames in row 3). The improvement is more substantial for challenging samples that the CAST originally fails on, increasing the mIoU from 69.5% to 76.6% (+7.1%) for samples with an original mIoU of less than 80%.
>
> [1] Sun et al. Test-Time Training with Self-Supervision for Generalization under Distribution Shifts. ICML 2020.\
> [2] Want et al. Tent: Fully Test-time Adaptation by Entropy Minimization. ICLR 2021.
>
> ---
> **[Q1] How to generate segments from ViT?**
>
> We generate segments from ViT by applying K-Means clustering to the final output tokens. We bilinearly upscale the feature map and then apply K-Means to the pixel-level features to align the segments with the input image resolution. Similar to CAST, we cross-reference clustering assignments across different levels to achieve hierarchical image segmentation. To maintain a consistent cluster hierarchy, we iteratively run K-Means, reducing the number of clusters by grouping the clusters from the previous iteration into coarser clusters. We have clarified this procedure on page 5 and in Appendix A.4 of our revised manuscript.
>
> ---
> **[Q2] PE added to the CLS token?**
>
> Following MoCo-v3, we used fixed sine-cosine positional embeddings for patches and an all-zero vector for the CLS token. The all-zero vector matches the length of positional embeddings with tokens but has no impact. We have clarified this in the equation on page 4 of our revised manuscript.
>
> ---
> Please let us know if you have any further concerns.
>
> Sincerely,\
> Authors

---

> > ### Comment · Reviewer_5FXt · 2023-11-22
> > **Thank you for your detailed response!**
> >
> > I appreciate the time taken to address my and the other reviewers' concerns.
> >
> > I think the TTA results are really nice and happy these were tried out. I also appreciate the reorganized table 3 - it's definitely clearer now.
> > I have raised my score to 8 accordingly.
> >
> > Thanks again!

---

### Author Response · Authors · 2023-11-19
**General Response**

Dear reviewers and AC,

We sincerely appreciate the time and effort on reviewing our manuscript.

As highlighted by the reviewers, our paper has a strong motivation with a simple and effective method, supported by comprehensive experiments.

We incorporated your comments into our revised manuscript, highlighting the updates in red. In particular, we included additional experiments and discussions:
- Extension to Test-time Adaptation (Appx. D)
- Ablation Study on Superpixel Methods (Appx. B.8)
- Additional Comparison with SOTA Token Pooling Methods (Table 6a)
- Additional Results on Inference Latency (Appx. B.1)

We believe our paper is strengthened by your constructive feedback and would be a valuable contribution to the ICLR community.

Sincerely,\
Authors

---

### Meta-Review · Area_Chair_rNGJ · 2023-12-05

**Metareview:**

This paper proposes an hierarchical alternative to patch-based representations for vision transformers by integrating superpixels and graph pooling into a transformer-based vision model. The resulting model, CAST, jointly learns hierarchical image segmentation while optimizing for an image recognition objective. Several advantages over regular vision transformers are demonstrated in the paper (improved segmentation, computational efficiency, and more interpretable attention maps).

All the reviewers agree that this is a high-quality paper with a solid motivation, great execution, elegant method, and convincing results. The method is immediately applicable to a range of vision problems and hence of high relevance to the community.

The authors have done a great job in addressing remaining concerns of the reviewers during the rebuttal stage. In particular the authors further demonstrated benefits compared to related work that use adaptive tokenization techniques.

Overall, this paper is a “clear accept” and a good candidate for either an oral or a spotlight presentation.

In terms of improvements for the final version of the paper: I recommend that the authors include (wall clock) runtime comparisons in their GFLOPS result tables, while also highlighting total #parameters for all models/baselines in Table 4.

**Justification For Why Not Higher Score:**

There are certainly some remaining weaknesses of the work (e.g. limited study on scalability to larger-scale models/datasets).

**Justification For Why Not Lower Score:**

Very high-quality paper for a topic which should be of broad interest to the ML / vision community.

---

### Decision · Program_Chairs · 2024-01-16

Accept (spotlight)